# Rethinking Feature Alignment in Generalist Graph Anomaly Detection: A Relational Fingerprint-based Approach

Yujing Liu [1]  Yixin Liu [1]  Yu Zheng [1]  Alan Wee-Chung Liew [1]  Xiaofeng Cao [2]  Shirui Pan [1]

## Abstract

Generalist graph anomaly detection (GAD) aims to detect anomalies on unseen graphs without graph-specific retraining. Nevertheless, existing approaches primarily focus on aligning heterogeneous features across different data domains via PCA-based projection, which harmonizes feature dimensions ignores feature semantics. As a result, GAD models fail to learn transferable semantic knowledge, and even exhibit negative transfer on unseen graphs. To address this issue, we propose a **Re**lational **Fi**ngerprint-based generalist **GAD** approach (REFI-GAD for short), aligning heterogeneous raw features with a universal and semantics-aware relational fingerprint (REFI) that encodes anomaly-indicative cues from both contextual and structural perspectives. Building on REFI, we design a fingerprint-grounded generalist GAD model, which combines a transformer-based encoder to capture domain-invariant knowledge with an SNR-guided refinement module for domain-specific adaptation. Extensive experiments on 14 datasets demonstrate that REFI-GAD significantly outperforms state-of-the-art methods. Code is available at https://github.com/Yujingcn/REFI-GAD-code.

## 1. Introduction

Graph Anomaly Detection (GAD) aims to identify anomalous nodes within graph data that significantly deviate from the majority distribution (Pan et al., 2026b; Liu et al., 2026; Zhao et al., 2025). This task is crucial for uncovering malicious or illicit behaviors in complex systems, and related techniques have played an indispensable role in real-world applications such as financial fraud detection (Wang et al.,

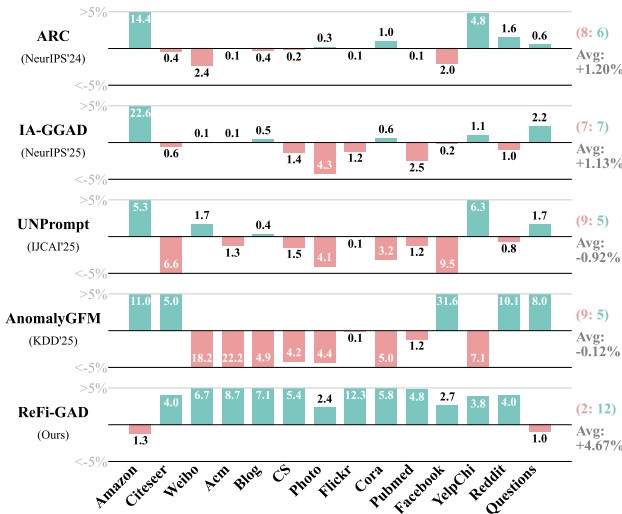

*Figure 1.* **Performance gain from pre-training**. Right: number of datasets with positive/negative transfer and average gain.

2019), network intrusion detection (Bilot et al., 2023), and social security monitoring (Liu et al., 2018). However, traditional GAD methods (Qiao et al., 2025b) heavily rely on dataset-specific training with full access to the target graph and its distribution, which tightly couples the detector to a fixed environment and limits its transferability to unseen graphs. This limitation hinders their practical deployment in privacy-sensitive or time-critical scenarios where collecting the target graph for training is costly or prohibited.

To break the dataset-specific training paradigm, recently, researchers have started to explore *generalist GAD* methods (Liu et al., 2024). Their key idea is to pre-train a one-for-all detector on a large and diverse collection of source graphs, so that the learned generalist detector can generalize to unseen graphs on-the-fly, without graph-specific retraining or fine-tuning (Niu et al., 2024; Qiao et al., 2025a; Zhang et al., 2026). Although they have shown promising performance under the generalist setting, we empirically show that their performance gains mainly arise from the architectural design of detection models, rather than transferable knowledge learned from large-scale source-graph pre-training. As shown in Figure 1, all four representative generalist approaches suffer from evident performance degradation on

[1]Griffith University, Gold Coast, Australia [2]Tongji University, Shanghai, China. Correspondence to: Shirui Pan <s.pan@griffith.edu.au>.

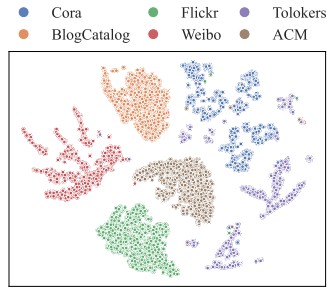
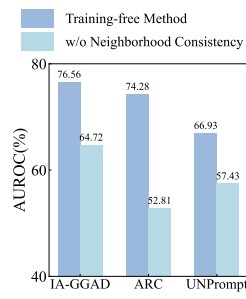

*(a)* PCA-aligned features      *(b)* Ablation performance

*Figure 2.* Analysis of generalist GAD methods, in terms of (a) aligned features visualized by t-SNE and (b) training-free performance w&w/o key design w.r.t. average AUROC over 14 datasets.

most datasets after source-domain training, with two of them even resulting in negative average improvements. This surprising observation indicates that **negative transfer** is widespread in the existing generalist GAD methods, exposing the limited transferability of the knowledge learned under the current cross-domain pre-training paradigm.

Despite the success of large-scale pre-training in other domains, it is less effective in generalist GAD. We attribute this failure to the unique nature of graph data: **feature heterogeneity** across domains. Unlike image data that shares a unified feature space, features in graph datasets can differ significantly in both dimensionality and semantics. For instance, node features in Cora are high-dimensional sparse bag-of-words vectors, while those in YelpChi are low-dimensional dense attributes (e.g., review-level statistics). To mitigate such heterogeneity, existing generalist GAD methods primarily rely on linear dimensionality reduction techniques (i.e., PCA (Abdi & Williams, 2010) or SVD (Stewart, 1993)) to align features across different datasets. However, as these techniques aim to maximally preserve the original feature distributions during dimensionality transformation, they inevitably inherit the semantic discrepancies between datasets. As visualized in Figure 2a, the features from different datasets still form clearly separated clusters in the aligned space, indicating that the alignment only matches feature dimensionality, but not feature semantics. In this case, a critical challenge in generalist GAD can be summarized as: ***how to construct transferable features across heterogeneous graph domains?***

Interestingly, the answer may already be hidden in plain sight: current generalist GAD methods already generalize well without training, implying that what truly transfers may lie in their architectural designs. Across these designs, a common design principle is to explicitly model neighborhood consistency. For example, ARC (Liu et al., 2024) introduces a residual encoder to capture the ego-neighbor discrepancy, while UNPrompt (Niu et al., 2024) leverages the similarity between a node embedding and its neighbor-

hood embedding as the anomaly score. If we remove these key modules, a significant drop is observed in the average performance of their training-free variants (see Figure 2b), which verifies that their generalization ability mainly stems from the built-in architectural priors that capture anomaly-indicative cues, such as neighborhood consistency. This leads to an intriguing question: ***can we explicitly extract these anomaly-indicative cues as transferable features to mitigate feature heterogeneity for generalist GAD?***

Motivated by this, we propose REFI-GAD, a novel approach that fundamentally reforms the feature alignment of generalist GAD. In REFI-GAD, we introduce Relational Fingerprint (REFI), an innovative transferable representation that replaces the heterogeneous raw features for generalist GAD feature alignment. Specifically, for diverse graph datasets, we uniformly extract five-dimensional relational attributes (i.e., REFI) for each node based on their raw features and topology, covering contextual patterns (neighborhood positional/directional consistency and global directional consistency) and structural patterns (local clustering coefficient and degree). Together, these attributes constitute a cross-domain universal fingerprint to encode the anomaly-indicative cues as transferable features. Building upon REFI, we design a fingerprint-grounded generalist GAD model, which jointly learns domain-shared representations and domain-specific adaptations for on-the-fly and adaptive anomaly detection on unseen graphs. Specifically, we design a transformer-based domain-shared encoder for capturing domain-invariant anomaly-related knowledge, followed by a Signal-to-Noise Ratio (SNR)-guided module for domain-adaptive refinement. Experimental results demonstrate that REFI-GAD not only achieves superior generalist detection performance, but also yields positive transfer on almost all datasets (see Figure 1). To sum up, the main contributions of this paper are summarized as follows:

- **Insight.** We rethink the feature alignment mechanism of existing generalist GAD methods and identify its inherent limitations, motivating a relational fingerprint (REFI) that captures transferable anomaly-indicative cues.
- **Method**. We propose REFI-GAD, a fingerprint-grounded generalist GAD framework that builds upon REFI to jointly model domain-shared representations and domain-specific adaptations for cross-graph anomaly detection.
- **Experiments.** Extensive experimental results on 14 real-world datasets verify the effectiveness, transferability, and scaling capability of REFI-GAD.

## 2. Related Works

### 2.1. Graph Anomaly Detection

Graph Anomaly Detection (GAD) aims to identify abnormal nodes, edges, or subgraphs that deviate from normal

graph patterns (Xi et al., 2025; Zhang et al., 2025; Zhao et al., 2026; Pan et al., 2026a). Existing methods are typically trained and evaluated within the same graph domain. Early approaches mainly adopt graph neural networks (GNNs), such as GCN (Kipf & Welling, 2016) and GAT (Veličković et al., 2017), with reconstruction- or classification-based objectives. Recent studies further explore anomaly-specific mechanisms. Contrastive methods, including CoLA (Liu et al., 2021) and ANEMONE (Jin et al., 2021), identify anomalies through inconsistencies between local and global representations. Spectral approaches such as BWGNN (Tang et al., 2022) capture anomalies via high-frequency graph signals, while generative methods like GGAD (Qiao et al., 2024) detect anomalies through reconstruction errors. Other methods, including SmoothGNN (Dong et al., 2025) and CHRN (Gao et al., 2023), enhance anomaly detection through feature smoothness and neighborhood consistency. Despite their effectiveness, these methods generally lack cross-domain generalization and often fail on unseen graph domains.

## 2.2. Generalist Graph Anomaly Detection

To improve cross-domain generalization, recent works investigate Generalist Graph Anomaly Detection (GAD) (Li et al., 2026; Pan et al., 2025), aiming to develop "one-for-all" models that generalize to unseen graphs without retraining. ARC (Liu et al., 2024) formulates GAD as a few-shot learning problem via in-context learning and smoothness-based alignment. AnomalyGFM (Qiao et al., 2025a) adopts a pre-train–fine-tune paradigm with prototype-based alignment for zero-shot inference and prompt tuning. UNPrompt (Niu et al., 2024) introduces unified neighborhood prompts to standardize heterogeneous graph information, while IA-GGAD (Zhang et al., 2026) learns domain-invariant features and structural correspondences for robust transferability. However, existing methods mainly focus on unifying feature dimensions or prompting strategies while overlooking semantic misalignment across heterogeneous graph domains, which often leads to negative transfer. A detailed related work is provided in Appendix A.

## 3. Preliminary

In this section, we introduce the notations and problem definition. A review of related work is provided in Appendix A.

**Notations.** Let $\mathcal{G} = (\mathcal{V}, \mathcal{E}, \mathbf{X})$ denote a graph, where $\mathcal{V} = \{v_1, \cdots, v_n\}$ is the set of $n$ nodes and $\mathcal{E} \subseteq \mathcal{V} \times \mathcal{V}$ represents the edges capturing pairwise relationships among nodes. Each node $v_i$ is endowed with a $d$-dimensional feature vector $\mathbf{x}_i \in \mathbb{R}^d$, and stacking all node features forms the feature matrix $\mathbf{X} \in \mathbb{R}^{n \times d}$. The graph topology is encoded by an adjacency matrix $\mathbf{A} \in \mathbb{R}^{n \times n}$, where $a_{ij} = 1$ if $(v_i, v_j) \in \mathcal{E}$ and $a_{ij} = 0$ otherwise. We denote $\mathbf{y} \in \mathbb{R}^n$

as the node label set, where $y_i = 0$ indicates that node $v_i$ is normal, while $y_i = 1$ corresponds to an anomalous node, Typically, $|\mathcal{V}_{norm}| \gg |\mathcal{V}_{ano}|$ in GAD problem.

**Generalist GAD Problem**. The goal of a generalist GAD is to learn a universal detection model, enabling a "one-model-for-all" anomaly detection across diverse graph domains (Qiao et al., 2025a; Liu et al., 2024). Formally, let $\mathcal{T}_s = \{\mathcal{D}_s^{(1)}, \cdots, \mathcal{D}_s^{(N)}\}$ denote the set of source-domain graphs used for training. The objective is to learn a generalist scoring function $f_\theta(\cdot)$ that can generalize to unseen target-domain graphs $\mathcal{T}_t = \{\mathcal{D}_t^{(1)}, \cdots, \mathcal{D}_t^{(N')}\}$. During inference, the model is provided with a few-shot support set containing $k$ normal and anomalous nodes as domain-specific references. Importantly, inference is performed in a training-free manner, where the support set is solely used to characterize contextual patterns without updating model parameters $\theta$. The goal is to produce an anomaly score such that $f_\theta(v_i) > f_\theta(v_j)$ for any anomalous node $v_i$ and normal node $v_j$ in the target dataset.

## 4. Methodology

In this section, we introduce REFI-GAD, a fingerprint-based generalist GAD approach. As shown in Figure 3, REFI-GAD mainly consists of two components: relational fingerprint extraction, which transforms heterogeneous features into universal relational fingerprint by capturing structural and contextual patterns, and fingerprint-grounded generalist GAD model, which learns anomaly discrimination from REFI via a domain-shared encoder and highlights discriminative dimensions with an SNR-based refinement strategy. Details are illustrated in the following subsections.

### 4.1. Relational Fingerprint Extraction

Due to the extreme heterogeneity in the dimension and semantics of raw features across domains (Ding et al., 2021), GAD models trained on source datasets are prone to negative transfer when generalized to target domains (Wang et al., 2023). To overcome this, we conduct a thorough analysis of current generalist GAD models and conclude that *their generalization ability mainly stems from built-in architectural inductive biases that capture anomaly-indicative cues*. Motivated by this, we propose a set of universal "Relational Fingerprints" (REFI) to characterize the relational attributes of nodes, which explicitly captures the anomaly-indicative cues from node features and graph structure. Specifically, we encode each node into a five-dimensional REFI vector, covering contextual patterns and structural patterns.

**Contextual Patterns.** In GAD tasks, under the homophily assumption (McPherson et al., 2001), feature consistency between a node and its neighborhood plays a central role in revealing anomalous behavior (Liu et al., 2021; Qiao

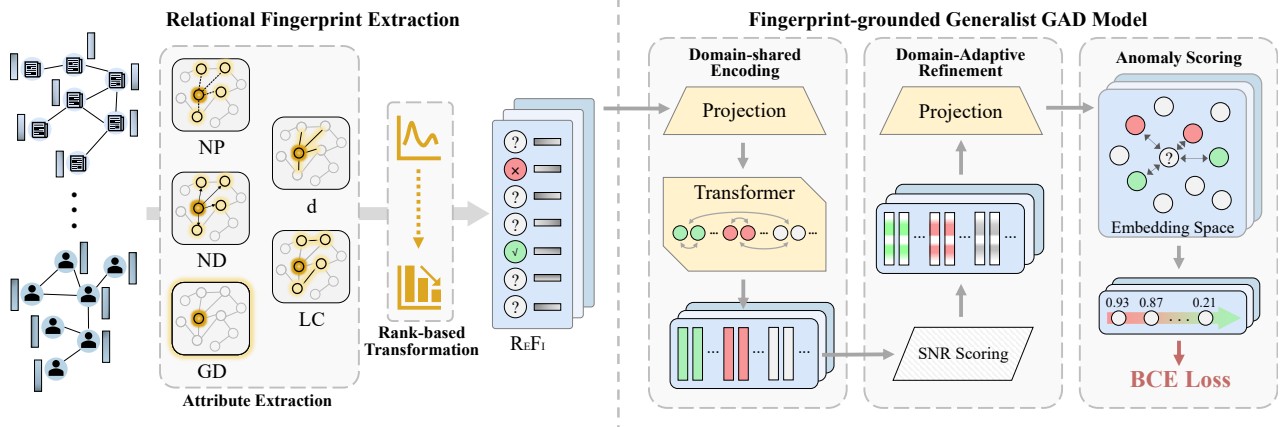

*Figure 3.* The framework of the proposed **REFI-GAD**. Firstly, the *relational fingerprint extraction* module derives five-dimensional REFI to capture universal anomaly-indicative patterns. Then, the *fingerprint-grounded generalist GAD model* learns domain-shared and domain-adaptive representations sequentially, followed by a distance-based scoring module to identify anomalies.

& Pang, 2023). Therefore, REFI first measures position and direction consistency with neighbors to capture local homophily. In addition, global directional consistency is introduced to capture a node's deviation from the global background, providing a more comprehensive contextual awareness from local to global.

***Dim❶: Neighborhood positional consistency.*** This metric quantifies the Euclidean distance between a node and its neighbors in the feature space. Usually, before evaluating neighborhood consistency, node features are propagated through graph convolution to construct a robust contextual representation. However, since anomalies are often embedded within normal communities, traditional graph convolution tends to cover up their deviations and thereby mask the anomalies. To address this, we design a similarity-aware graph convolution that re-weights edges according to raw feature similarity, defined as:

$$\bar{\mathbf{A}} = \hat{\mathbf{A}} \odot (\mathbf{X}\mathbf{X}^\top), \tag{1}$$

where $\odot$ denotes the Hadamard product, $\hat{\mathbf{A}} = \tilde{\mathbf{D}}^{-\frac{1}{2}} \tilde{\mathbf{A}} \tilde{\mathbf{D}}^{-\frac{1}{2}}$ where $\tilde{\mathbf{A}} = \mathbf{A} + \mathbf{I}$, $\tilde{\mathbf{D}}$ is a diagonal matrix with $\tilde{d}_{ii} = \sum_j \tilde{a}_{ij}$, and $\mathbf{I}$ is the identity matrix. Based on the re-weighted adjacency matrix $\bar{\mathbf{A}}$, we perform feature aggregation to obtain the robust node representation, i.e., $\bar{\mathbf{X}} \in \mathbb{R}^{n \times d} = \bar{\mathbf{A}}\mathbf{X}$. This similarity-aware convolution suppresses feature propagation from dissimilar neighbors, preserving anomalous patterns in the convolved representation $\bar{\mathbf{X}}$. Then, we calculate the neighborhood positional consistency for each node $v_i$ by quantifying its average Euclidean distance from its neighbors:

$$\text{NP}_i = \frac{1}{|\mathcal{N}_i|} \sum_{v_j \in \mathcal{N}_i} \|\bar{\mathbf{x}}_i - \bar{\mathbf{x}}_j\|_2, \tag{2}$$

where $\mathcal{N}_i$ is the set of neighbors for node $v_i$ and $\|\cdot\|_2$ denotes the L2-norm. Intuitively, a larger value of NP indicates

that the node is spatially distant from its local community, suggesting a lower positional consistency with its context.

***Dim❷: Neighborhood directional consistency.*** To provide a more comprehensive assessment of neighborhood homophily, we further investigate the directional consistency of node features with their neighbors. Specifically, this metric is calculated as the average cosine similarity between node $v_i$ and its neighbor set $\mathcal{N}_i$:

$$\text{ND}_i = \frac{1}{|\mathcal{N}_i|} \sum_{v_j \in \mathcal{N}_i} \frac{\bar{\mathbf{x}}_i \cdot \bar{\mathbf{x}}_j}{\|\bar{\mathbf{x}}_i\|_2 \|\bar{\mathbf{x}}_j\|_2}. \tag{3}$$

A smaller ND indicates that the semantic direction of the node deviates from the consensus of its local community, suggesting a lower degree of homophily.

***Dim❸: Global directional consistency.*** In addition to local consistency, we also incorporate a global perspective to detect statistical outliers that may violate the dominant semantic trend of the entire graph. From a global view, we aim to capture the dominant data manifold. Therefore, we utilize conventional graph convolution to aggregate second-order contextual information. The node representation is computed as $\hat{\mathbf{X}} \in \mathbb{R}^{n \times d} = \hat{\mathbf{A}}^2\mathbf{X}$. Subsequently, we compute the global directional consistency (GD) as:

$$\text{GD}_i = \frac{\hat{\mathbf{x}}_i \cdot \mathbf{c}_g}{\|\hat{\mathbf{x}}_i\|_2 \|\mathbf{c}_g\|_2}, \quad \text{with } \mathbf{c}_g = \frac{1}{n} \sum_{k=1}^n \hat{\mathbf{x}}_k, \tag{4}$$

where $\mathbf{c}_g$ is the global distribution center. A lower GD suggests that the node's semantic orientation deviates significantly from the overall data manifold, marking it as a potential global outlier.

**Structural Patterns.** In addition to contextual patterns, structural patterns provide complementary cues for anomaly

detection, particularly under feature camouflage or structural attack scenarios where anomalies imitate normal attributes (Akoglu et al., 2010). In such cases, relying solely on feature consistency may overlook nodes with irregular topological connections. Therefore, we explicitly incorporate structural properties into the relational fingerprint. Specifically, we extract degree and local clustering coefficient to characterize anomalies in terms of connectivity prominence and community tightness.

***Dim❹: Degree.*** The degree of a node serves as a fundamental measure of its connectivity prominence. We calculate the degree $d_i$ of node $v_i$ based on the adjacency matrix $\mathbf{A}$:

$$\mathbf{d}_i = \sum_{j=1}^{n} \mathbf{a}_{ij}. \tag{5}$$

In specific domains, anomalies may manifest as statistical extremes in connectivity. For instance, in fraud detection, spammers or attackers typically appear as "hubs" with abnormally high degrees, connecting to numerous targets, whereas other outliers might appear as isolated nodes with sparse connections.

***Dim❺: Local clustering coefficient.*** This metric quantifies the tightness of the local community structure surrounding a node. It measures the tendency of a node's neighbors to form a connected clique. Formally, let $T_i$ denote the number of triangles (closed triplets) involving node $v_i$. The clustering coefficient is defined as:

$$\mathrm{LC}_i = \frac{2T_i}{d_i(d_i - 1)}, \tag{6}$$

where $d_i$ is the degree of node $v_i$. A high LC implies that the node is embedded in a dense, cohesive community.

**Relational Fingerprint Construction.** Different datasets often exhibit distinct statistical characteristics (e.g., degree distributions), resulting in cross-domain variations in the raw values of these metrics. To mitigate this domain gap and align relational fingerprints within a unified distribution space, we adopt a rank-based transformation strategy for REFI construction. This strategy converts absolute numerical values into relative percentile ranks, ensuring the scale unification across datasets. Specifically, the rank-based transformation is defined as $r(m_i) = \mathrm{rank}(m_i)/n$, where $m \in \{\mathrm{NP}, \mathrm{ND}, \mathrm{GD}, \mathrm{d}, \mathrm{LC}\}$ denotes the relational attribute, and $\mathrm{rank}(m_i)$ is the ascending rank index of node $v_i$ within the dataset. This transformation strategy unifies attribute scales while effectively circumventing the feature collapse caused by traditional normalization methods.

Finally, we integrate these rank-transformed metrics to construct the REFI matrix $\mathbf{P}$ for a graph, where each row $\mathbf{p}_i$ is the fingerprint for each node $v_i \in \mathcal{V}$:

$$\mathbf{p}_i = \big[r(\mathrm{NP}_i), r(\mathrm{ND}_i), r(\mathrm{GD}_i), r(\mathrm{d}_i), r(\mathrm{LC}_i)\big]. \tag{7}$$

This compact vector $\mathbf{p}_i$ comprehensively encodes the node's anomaly-indicative cues, capturing deviations across local geometry, semantic directions, global trends, and topological structures. By mapping heterogeneous raw features to the unified REFI space, we achieve feature and semantic alignment of diverse graph data, providing a solid foundation for cross-domain knowledge transfer.

### 4.2. Fingerprint-grounded Generalist GAD Model

In real-world graph data, anomalous behaviors are typically manifested through the joint interplay of multi-dimensional relational attributes. Accordingly, after constructing REFI, we further employ a deep model to capture the shared latent correlations among relational dimensions for generalist anomaly detection. Furthermore, to handle domain-specific pattern variations, we propose a domain-adaptive dimension focusing strategy, which further enhances the model's cross-domain generalization capability.

**Domain-Shared Representation Encoding.** To effectively learn the dependencies among different relational attributes, we introduce a domain-shared encoder that projects relational fingerprints into a high-dimensional latent space to capture complex non-linear interactions across attributes in REFI. Specifically, this encoder consists of a feature projector and a context-aware Transformer.

First, we employ a feature projector (implemented via a single-layer MLP (Chen et al., 2020)) to map the 5-dimensional REFI $\mathbf{P}$ into a high-dimensional representation space, denoted as $\mathbf{H}^{(0)} = \mathrm{MLP}(\mathbf{P})$. Subsequently, considering that the complex dependencies among relational dimensions may vary across domains, we incorporate domain-specific contextual knowledge when modeling these latent interactions. We employ a Context-aware Transformer (Vaswani et al., 2017) to capture these intricate dependencies, leveraging the few-shot support samples to provide domain-specific contextual information. Specifically, we construct a unified input sequence by concatenating the embeddings of the support and query sets. Formally, the input sequence $\mathbf{Z}^{(0)}$ is defined as:

$$\mathbf{Z}^{(0)} = \mathrm{Concat}\big(\mathbf{H}^{(0)}_{[\mathcal{S}_n]}, \mathbf{H}^{(0)}_{[\mathcal{S}_a]}, \mathbf{H}^{(0)}_{[\mathcal{Q}]}\big) \in \mathbb{R}^{(2k+n_b) \times d'}, \tag{8}$$

where $\mathcal{S}_n$ and $\mathcal{S}_a$ represent the indices of normal and anomalous samples within the support sets, $\mathcal{Q}$ denotes the batch samples within query set, $k$ is the size of each support set, $d'$ represents the hidden dimension, and $n_b$ is the batch size of query samples.

Then, to guide the model in encoding query node representations with domain contextual knowledge , we design an attention mask to regulate information flow within the Transformer. Specifically, all support nodes are fully visible to each other while isolated from the query nodes, establishing a stable global contextual background. Mean-

while, query nodes attend only to the support set, thereby capturing latent dependencies conditioned on the domain-specific background. Formally, the attention mask $\mathbf{M} \in \mathbb{R}^{(2k+n_b) \times (2k+n_b)}$ is defined as:

$$m_{ij} = \begin{cases} 0, & \text{if } j \leq 2k \text{ or } i = j, \\ -\infty, & \text{otherwise}. \end{cases} \quad (9)$$

where indices $j \leq 2k$ correspond to the support set positions. Finally, we feed the input sequence $\mathbf{Z}^{(0)}$ along with the attention mask $\mathbf{M}$ into the transformer encoder to perform context-aware feature extraction:

$$\mathbf{Z}^{(l+1)} = \text{FFN}\left(\text{Softmax}\left(\frac{\mathbf{Q}\mathbf{K}^\top}{\sqrt{d'}} + \mathbf{M}\right)\mathbf{V}\right), \quad (10)$$

where $\mathbf{Q}, \mathbf{K}, \mathbf{V}$ are the query, key, and value matrices, computed by $\mathbf{Q} = \mathbf{Z}^{(l)}\mathbf{W}_Q^{(l)}$, $\mathbf{K} = \mathbf{Z}^{(l)}\mathbf{W}_K^{(l)}$, and $\mathbf{V} = \mathbf{Z}^{(l)}\mathbf{W}_V^{(l)}$, with each $\mathbf{W}$ as learnable parameters. After $L$ layers, we obtain the node representation $\mathbf{H}^{(1)} = \mathbf{Z}^{(L)}$, which effectively encodes the complex interactions among different relational attributes, thereby enhancing the discriminability between normal and anomalous nodes.

**SNR-guided Domain-Adaptive Refinement.** Although the domain-shared encoder captures interactions among relational attributes, their contributions to anomaly detection can vary across domains. To address this, we assess the discriminative importance of each representation dimension based on the domain-specific context distribution, which highlights the domain-specific key dimensions for domain-adaptive anomaly detection.

Specifically, we first estimate the global normal and anomalous centers based on the support and query sets:

$$\mathbf{h}_n = \frac{1}{k + n_b}\left(\sum_{i \in \mathcal{S}_n} \mathbf{h}_i^{(1)} + \sum_{j \in \mathcal{Q}} \mathbf{h}_j^{(1)}\right), \quad \mathbf{h}_a = \frac{1}{k}\sum_{i \in \mathcal{S}_a} \mathbf{h}_i^{(1)}, \quad (11)$$

where $\mathbf{h}_n$ and $\mathbf{h}_a$ denote the global normal and anomalous centers, respectively, representing the domain-specific distributions in the target dataset. Then, we compute a dimension-wise Signal-to-Noise Ratio (SNR) (Golub et al., 1999) to quantify the discriminative importance of each representation dimension:

$$\mathbf{s} = \frac{(\mathbf{h}_a - \mathbf{h}_n)^2}{\boldsymbol{\sigma}_n^2 + \epsilon}, \quad (12)$$

where $\mathbf{s} \in \mathbb{R}^{d'}$, $\boldsymbol{\sigma}_n^2 = \text{Var}(\mathcal{S}_n \cup \mathcal{Q})$ denotes the element-wise variance of the normal background distribution, and $\epsilon$ is a small constant for numerical stability. Based on the SNR scores, we generate adaptive dimension weights via $\mathbf{m} = \sigma(\lambda \cdot \mathbf{s} + \beta)$, with $\lambda$ and $\beta$ as learnable scaling parameters and $\sigma(\cdot)$ as sigmoid function.

Finally, we recalibrate the node representations extracted by the domain-shared encoder using the dimension weights, followed by a linear projection layer to obtain the final node representations $\mathbf{H} \in \mathbb{R}^{(2k+n_b) \times d'}$, denoted as $\mathbf{H} = \mathbf{W}'(\mathbf{H}^{(1)} \odot \mathbf{m})$, where $\mathbf{W}'$ is learnable parameter.

**Anomaly Scoring.** Given the final node representations $\mathbf{H}$, we perform anomaly scoring by measuring the relative similarity of query nodes to the normal and anomalous support sets in the embedding space. Concretely, for each query node, we compute its pairwise cosine similarity with all support nodes:

$$s_{i,j} = \frac{\mathbf{h}_i^\top \mathbf{h}_j}{\|\mathbf{h}_i\|_2 \|\mathbf{h}_j\|_2}, \quad \forall n_i \in \mathcal{Q}, n_j \in \mathcal{S}, \quad (13)$$

where $\mathcal{S} = \mathcal{S}_n \cup \mathcal{S}_a$ is entire support set and $s_{i,j}$ denotes the similarity score between query node $n_i$ and support node $n_j$. Then, a learnable temperature parameter $\tau$ is used to rescale the similarity scores, which are then normalized via a Softmax to produce anomaly weights over the support set:

$$\alpha_{i,j} = \frac{\exp(\tau \cdot s_{i,j})}{\sum_{n_k \in \mathcal{S}} \exp(\tau \cdot s_{i,k})}. \quad (14)$$

The anomaly score for query node $v_i$ is then computed as:

$$\tilde{y}_i = \frac{1}{2}\left(\sum_{n_j \in \mathcal{S}_a} \alpha_{i,j} - \sum_{n_j \in \mathcal{S}_n} \alpha_{i,j} + 1\right), \quad (15)$$

where $\tilde{y}_i \in [0, 1]$ indicates the likelihood of query node $n_i$ being anomalous.

**Model Training.** During training on the source datasets, we optimize the model by minimizing the binary cross-entropy (BCE) loss between the predicted anomaly scores $\tilde{y}_i$ and the ground-truth labels $y_i \in \{0, 1\}$ of query nodes:

$$\mathcal{L} = -\frac{1}{|\mathcal{Q}|}\sum_{i \in \mathcal{Q}}\left[y_i \log \tilde{y}_i + (1 - y_i)\log(1 - \tilde{y}_i)\right]. \quad (16)$$

The framework is trained in an end-to-end episodic manner, enabling the model to capture transferable anomaly detection patterns that generalize to unseen target domains. Detailed algorithmic descriptions and complexity analysis are provided in Appendix B and C, respectively.

# 5. Experiments

## 5.1. Experimental Setup

**Datasets.** To evaluate the generalization capability of the models, we partition the datasets into two disjoint groups and train all comparative methods on one group of graph datasets and test them on the other. To mitigate domain bias, we evenly distribute datasets from diverse domains (including citation , social , e-commerce , and

*Table 1.* Anomaly detection performance in terms of AUROC (%). Highlighted are the results ranked first, second, and third.

| Method | Group 1 (Models Trained on Group 2) | | | | | | | Group 2 (Models Trained on Group 1) | | | | | | | Rank |
|---|---|---|---|---|---|---|---|---|---|---|---|---|---|---|---|
| | Cite | CS | ACM | Blog | Amz | Photo | Weibo | Cora | Pubmed | Flickr | FB | Yelp | Quest | Reddit | |
| GAD Methods | | | | | | | | | | | | | | | |
| GCN | 48.32 | 56.19 | 50.43 | 43.06 | 58.69 | 47.70 | 46.40 | 32.41 | 33.68 | 38.07 | 76.35 | 51.21 | 41.37 | 46.80 | 11.86 |
| GAT | 63.11 | 59.25 | 61.48 | 60.50 | 54.84 | 45.63 | 73.51 | 56.07 | 67.21 | 56.05 | 55.33 | 50.43 | 57.37 | 43.39 | 9.93 |
| CoLA | 73.81 | 65.99 | 54.94 | 60.58 | 65.02 | 65.18 | 41.08 | 66.02 | 70.09 | 60.64 | 70.88 | 52.41 | 53.23 | 50.60 | 8.29 |
| SmoothGNN | 90.42 | 78.65 | 78.00 | 73.06 | 50.75 | 44.80 | 85.96 | 88.39 | 78.22 | 76.95 | 53.62 | 61.30 | 59.01 | 54.09 | 5.43 |
| ANEMONE | 41.36 | 42.18 | 45.53 | 35.18 | 43.51 | 49.22 | 35.57 | 47.08 | 37.65 | 37.07 | 45.55 | 50.88 | 51.55 | 52.41 | 13.21 |
| AHFAN | 48.02 | 59.09 | 63.70 | 59.60 | 29.50 | 52.16 | 63.03 | 55.89 | 64.08 | 62.77 | 32.11 | 49.76 | 54.01 | 46.90 | 11.07 |
| BWGNN | 60.91 | 59.12 | 61.80 | 67.26 | 50.40 | 60.82 | 81.62 | 63.62 | 68.19 | 68.37 | 41.55 | 52.76 | 57.59 | 60.24 | 7.64 |
| BGNN | 55.14 | 52.14 | 52.01 | 52.65 | 52.17 | 47.26 | 77.47 | 52.66 | 58.46 | 52.56 | 65.80 | 50.67 | 56.80 | 41.80 | 11.07 |
| CHRN | 74.25 | 78.01 | 76.88 | 65.09 | 52.93 | 51.99 | 57.77 | 68.18 | 75.15 | 64.32 | 33.07 | 51.72 | 59.81 | 53.45 | 7.29 |
| GGAD | 67.03 | 64.76 | 67.02 | 58.83 | 50.36 | 58.72 | 70.23 | 71.15 | 69.82 | 59.79 | 20.24 | 52.14 | 58.92 | 53.78 | 8.57 |
| Generalist GAD Methods | | | | | | | | | | | | | | | |
| ARC | 91.59 | 82.76 | 79.87 | 74.07 | 79.08 | 75.87 | 85.54 | 87.91 | 85.62 | 74.53 | 67.69 | 53.34 | 57.96 | 60.94 | 3.50 |
| AnomalyGFM | 50.44 | 43.47 | 36.98 | 45.16 | 48.62 | 44.86 | 43.16 | 43.94 | 48.96 | 47.67 | 71.08 | 46.21 | 53.34 | 54.71 | 12.21 |
| UNPrompt | 71.86 | 74.65 | 73.92 | 68.94 | 65.08 | 68.68 | 47.21 | 69.56 | 82.72 | 69.60 | 75.82 | 53.91 | 46.72 | 55.51 | 5.93 |
| IA-GGAD | 91.74 | 92.94 | 90.98 | 75.22 | 82.49 | 69.45 | 91.70 | 88.08 | 88.07 | 67.86 | 79.76 | 52.45 | 59.13 | 57.76 | 2.86 |
| REFI-GAD | 96.62 | 98.53 | 95.54 | 76.79 | 71.21 | 75.95 | 92.66 | 96.14 | 97.66 | 89.61 | 94.67 | 83.40 | 60.27 | 62.11 | 1.14 |

behavioral/interaction networks) across the two groups. Specifically, **Group 1** consists of Citeseer, CS, ACM, BlogCatalog, Amazon, Photo, and Weibo, while **Group 2** comprises Cora, Pubmed, Flickr, Facebook, YelpChi, Questions, and Reddit. Comprehensive dataset details are provided in Appendix D.1.

**Comparison Methods**. We compare REFI-GAD with the up-to-date and state-of-the-art methods. The comparison methods include two classical baselines (GCN (Kipf & Welling, 2016) and GAT (Veličković et al., 2017)), three self-supervised GAD methods (CoLA (Liu et al., 2021), SmoothGNN (Dong et al., 2025), and ANEMONE (Jin et al., 2021)), five supervised GAD methods (AHFAN (Wang et al., 2025), BWGNN (Tang et al., 2022), BGNN (Ivanov & Prokhorenkova, 2021), CHRN (Gao et al., 2023), and GGAD (Qiao et al., 2024)) and four generalist GAD methods (ARC (Liu et al., 2024), AnomalyGFM (Qiao et al., 2025a), UNPrompt (Niu et al., 2024), and IA-GGAD (Zhang et al., 2026)). Detailed descriptions of these methods are provided in Appendix D.2.

**Evaluation and Implementation.** Following existing work (Tang et al., 2023; Qiao & Pang, 2023; Pang et al., 2021), we adopt AUROC and AUPRC as evaluation metrics. To ensure the reliability of the results, we perform five-fold cross-validation and report both the mean and standard deviation. . In our experiments, all comparison methods are trained on one group of datasets and tested on the other, with the two groups alternating as training and testing sets. For our method, the default value of $k$ is 50, and the support set size is kept consistent across all few-shot methods. Furthermore, for all traditional GAD algorithms, we apply PCA to unify the feature dimensions of the datasets, enabling

evaluation in a generalist GAD scenario. Since our goal is to train generalist GAD models, we do not perform dataset-specific hyperparameter tuning, and instead use the same set of hyperparameters across all testing datasets. Further experimental details can be found in the Appendix D.3.

**5.2. Performance Comparison**

We evaluate the effectiveness of REFI-GAD by comparing it with all competing methods on 14 datasets. Table 1 reports the average AUROC of different algorithms, and more experimental results are provided in Appendix E.1. We make the following observations. ❶ Our method achieves SOTA performance on almost all datasets. Specifically, REFI-GAD yields an average improvement of 7.39% over the strongest baseline, IA-GGAD, and outperforms the weakest baseline, ANEMONE, by an average of 41.17% across all datasets. This is attributed to our proposed feature alignment and feature extraction strategies, which enable the model to capture domain-invariant relational characteristics and effectively extract discriminative node embeddings in unseen target domains. ❷ REFI-GAD significantly outperforms traditional GAD methods. Compared with SmoothGNN, the best-performing conventional GAD approach, REFI-GAD achieves an average improvement of 15.57%. This is because the proposed dimension focusing strategy enables the model to more effectively identify anomaly-relevant features in the target domain, thereby achieving better anomaly detection performance. ❸ REFI-GAD also demonstrates clear advantages over existing generalist GAD models, achieving an average improvement of over 7.39% across all datasets. The reason is that the proposed attribute fingerprint construction effectively achieves feature alignment across datasets, thereby providing a solid foundation for the model's cross-

*Table 2.* AUROC (%) of REFI-GAD and its variants.

| Variant | Cora | Pubmed | Flickr | FB | Yelp | Quest | Reddit |
|---|---|---|---|---|---|---|---|
| w/o R&D | 52.80 | 66.75 | 73.41 | 55.04 | 55.55 | 55.23 | 58.27 |
| w/o R | 53.28 | 66.26 | 71.71 | 54.88 | 54.15 | 56.24 | 58.66 |
| w/o D | 96.07 | 97.33 | 88.08 | 93.30 | 81.29 | 60.16 | 61.48 |
| REFI-GAD | **96.14** | **97.66** | **89.61** | **94.67** | **83.40** | **60.27** | **62.11** |

*Table 3.* AUROC (%) of REFI-GAD without different REFI.

| Dataset | w/o LC | w/o DE | w/o NP | w/o ND | w/o GD | REFI-GAD |
|---|---|---|---|---|---|---|
| Cora | 94.70 | 96.08 | 94.23 | 93.93 | **96.62** | 96.14 |
| Pubmed | 96.82 | 96.56 | 95.12 | 97.12 | 97.62 | **97.66** |
| Flickr | 88.23 | 79.78 | 89.53 | 85.85 | 82.05 | **89.61** |
| Facebook | 92.59 | 88.65 | 94.92 | **95.96** | 88.56 | 94.67 |
| YelpChi | 79.28 | 81.44 | 81.93 | 82.14 | 80.37 | **83.40** |
| Questions | 60.46 | **61.62** | 60.05 | 59.70 | 58.93 | 60.27 |
| Reddit | 59.74 | 60.89 | 62.02 | 60.53 | 59.46 | **62.11** |
| Average | 81.69 | 80.72 | 82.54 | 82.18 | 80.52 | **83.41** |

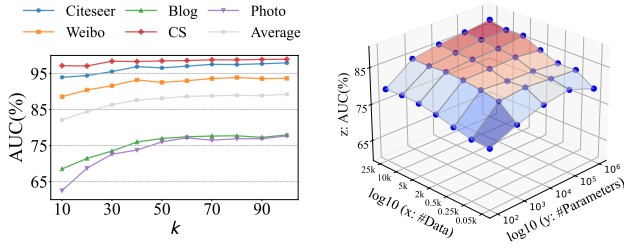

*(a)* Sensitivity Analysis of $k$     *(b)* Generalization Analysis

*Figure 4.* Context Size Sensitivity and Generalization Analysis.

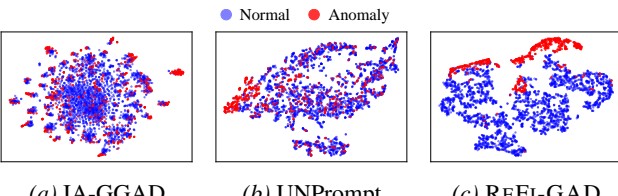

*(a)* IA-GGAD     *(b)* UNPrompt     *(c)* REFI-GAD

*Figure 5.* visualization on the target dataset Pubmed.

domain generalization.

### 5.3. Ablation Study of Key Designs

REFI-GAD comprises two key designs: relational fingerprint construction and SNR-guided domain-adaptive refinement. To evaluate the contribution of each design to the overall performance, we conducted a systematic ablation study. Specifically, in the **w/o R** setting, we employ PCA for feature alignment, while in the **w/o D** setting, we remove the domain-adaptive refinement module from the model. The experimental results are reported in Table 2. We have the following finding:

❶ Compared to the **w/o R** variant, the complete REFI-GAD model achieves an average improvement of 20.39% across all datasets. This demonstrates that our proposed relational fingerprint construction realizes cross-dataset feature alignment more effectively than a simple PCA strategy, providing a solid foundation for cross-domain generalization and leading to significant performance gains. ❷ Compared to **w/o D** variant, the complete REFI-GAD yields an average improvement of 0.70% across all datasets. This indicates that our designed domain-adaptive dimension effectively identifies anomaly-related dimensions within the target domain, thereby further enhancing model performance.

### 5.4. Relational Fingerprint Effectiveness Analysis

To investigate the necessity of each attribute within REFI, we conduct an ablation study by individually removing each dimension. The results are reported in Table 3. It can be clearly observed that all five attributes are indispensable, as removing any single attribute consistently degrades the overall performance of REFI-GAD by 0.87% to 2.89%. This is

primarily because the five relational attributes characterize node anomalies from complementary perspectives. Consequently, excluding any attribute inevitably results in the loss of critical anomaly-related signals, ultimately leading to inferior detection performance.

### 5.5. Effectiveness of #Context Nodes

We investigate the sensitivity of REFI-GAD to the context size $k$ by vary the value of $k$ within the range $\{10, 20, \ldots, 100\}$. As shown in Figure 4a, the model performance improves as $k$ increases, and this trend begins to level off around $k = 50$. This is because when $k$ is too small, the model struggles to obtain an accurate background distribution from the support set, resulting in slightly degraded quality of the extracted node embeddings and, consequently, reduced anomaly detection performance.

### 5.6. Model Generalization Ability Analysis

To verify the scaling capability of REFI-GAD as a foundation model, we investigated the relationship between model performance (AUROC) and two key variables: training data size and model parameters. Specifically, we scaled the training data size from 0.05k to 25k and the model parameters from $10^2$ to $10^6$. The experimental results are illustrated in Figure 4b. As shown in the figure, the performance of REFI-GAD improves progressively with the increase of both model capacity and training samples. Notably, when the model parameters reach the million scale ($10^6$), a larger volume of training samples (2k) is required to effectively fit the model and unlock its potential. Consequently, the optimal performance is achieved in the region characterized by "large data size and large parameter size" (the top red

*Table 4.* Anomaly Detection Performance under Fully Cross-Domain Transfer (AUROC (%))

| Method | Cora | Citeseer | ACM | CS | Pubmed |
|---|---|---|---|---|---|
| ARC | 88.14 | 90.89 | 79.80 | 82.53 | 85.60 |
| AnomalyGFM | 39.19 | 43.14 | 38.53 | 39.23 | 37.35 |
| UNPrompt | 59.35 | 67.26 | 72.47 | 73.60 | 77.81 |
| IA-GGAD | 88.23 | 91.97 | 90.77 | 92.70 | 89.10 |
| REFI-GAD | **95.66** | **95.33** | **93.01** | **97.14** | **95.44** |

area in the figure), where the AUROC exceeds 87%. These results demonstrate that REFI-GAD follows the scaling law observed in general machine learning, confirming its potential as a foundation model capable of adapting to complex, large-scale real-world graph scenarios through scaling.

### 5.7. Analysis of fully Cross-Domain Transferability

To further investigate the generalization capability of the model under the fully cross-domain transfer scenario, we conduct comparative experiments between existing generalist GAD models and REFI-GAD. Specifically, we treat the citation domain as the target domain and use the remaining three domains as source domains for model training. The experimental results are summarized in Table 4. Furthermore, we analyze the performance gains obtained by these methods through source-domain training in this scenario, with results illustrated in Figure 6.

We make the following observations. ❶ REFI-GAD maintains superior anomaly detection performance even in the fully cross-domain setting. Specifically, compared with IA-GGAD, the best-performing baseline among existing generalist GAD models, our method achieves an average improvement of 4.76% across the five datasets in the target domain. ❷ Existing approaches generally suffer from negative transfer in full cross-domain scenario. Their source-domain training often leads to negative average improvements on the target domain. In stark contrast, REFI-GAD achieves an average performance gain of 4.15% through source-domain training. We attribute this success to the proposed Relational Fingerprints (REFI), which effectively unify feature semantics across heterogeneous domains. This mechanism enables the model to capture generic, cross-domain transferable knowledge (e.g., latent correlations among relational dimensions), thereby endowing the model with genuine cross-domain generalization capabilities.

### 5.8. Visualization Experiments

To intuitively compare the cross-domain transfer performance between our method and existing generalist models, we visualized the node embeddings on the target domain, following the setup of the comparative experiments, as shown in Figure 5 (additional comparisons are provided in the Ap-

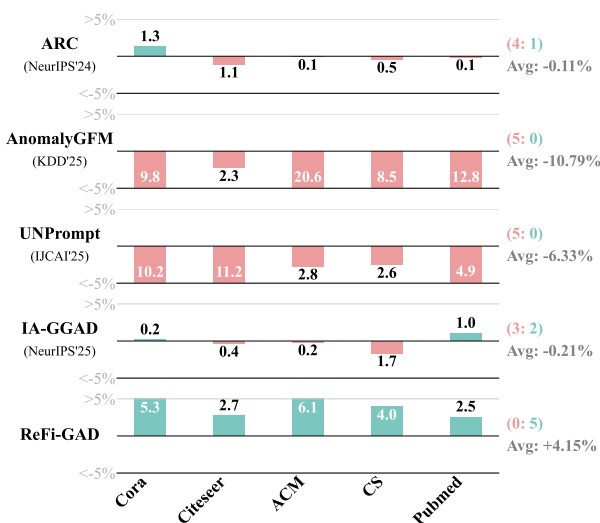

*Figure 6.* **Performance gain from source-domain training.** Right: number of datasets with positive/negative transfer and average gain.

pendix E.3). As illustrated, the embedding distributions of existing models on the target domain are relatively entangled, making it difficult to effectively separate normal and anomalous nodes. In contrast, the node embeddings produced by our method exhibit significantly improved anomaly separability. This advantage can be attributed to two key factors. First, the proposed relational fingerprint construction enables more effective cross-dataset feature alignment, thereby enhancing the model's cross-domain transferability. Second, the introduced domain-adaptive dimension focusing module adaptively identifies feature dimensions that are highly relevant to anomalies, further increasing the separation between normal and anomalous nodes in the embedding space.

## 6. Conclusion

In this paper, we propose a novel generalist GAD method, REFI-GAD, aiming to align heterogeneous graph features in a new manner. We introduce relational fingerprints (REFI), a semantics-aware transferable representation to capture anomaly-indicative cues across heterogeneous graph domains. On top of REFI, we build a fingerprint-grounded GAD framework that models transferable anomaly-related knowledge and supports domain-specific adaptation. Extensive experiments on 14 benchmark datasets demonstrate that REFI-GAD achieves strong detection performance, positive transferability, and scaling capability. The **limitations** of REFI-GAD include the requirement of contextual samples for reliable anomaly scoring, and the need for manual design in constructing REFI. Therefore, promising **future directions** lie in fingerprint-based zero-shot generalist GAD and automated relational fingerprint learning.

## Acknowledgements

The work of S. Pan was partially supported by the Australian Research Council (ARC) under Grant Nos. DP240101547 and FT210100097. The work of Y. Liu was partially supported by the ARC under Grant No. DE260101172.

## Impact Statement

This paper presents methodological advances for generalist Graph Anomaly Detection (GAD), aiming to improve model generalization across diverse graph domains. As a general-purpose learning framework, the proposed approach may support a wide range of applications involving graph-structured data, such as social network analysis, financial risk monitoring, and system reliability assessment. While progress in anomaly detection may have broader societal implications, ranging from ethical considerations in surveillance to the robustness of safety-critical systems, we do not identify any specific negative consequences that warrant particular emphasis at this time. However, we encourage further discussion and evaluation of the broader impact, particularly regarding fairness and privacy, as the field continues to evolve.

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

# A. Related Works

## A.1. Graph Anomaly Detection

Graph Anomaly Detection (GAD) seeks to identify nodes, edges, or subgraphs that significantly deviate from typical graph patterns. Traditional GAD methods are largely domain-specific, trained and evaluated on the same graph. Early approaches typically employ graph neural networks (GNNs) such as GCN (Kipf & Welling, 2016) and GAT (Veličković et al., 2017) as backbones, with reconstruction- or classification-based objectives.

To better capture anomaly-specific properties, recent works introduce specialized mechanisms. For example, contrastive learning frameworks like CoLA (Liu et al., 2021) and ANEMONE (Jin et al., 2021) detect anomalies by maximizing agreement between local subgraphs and global graph representations, treating inconsistent nodes as outliers. Recognizing that anomalies often manifest as high-frequency noise in the spectral domain, BWGNN (Tang et al., 2022) applies beta-wavelet filters to capture spectral anomalies. Similarly, generative models such as GGAD (Qiao et al., 2024) employ GANs to model normal node distributions and detect anomalies via reconstruction errors. Other approaches, including SmoothGNN (Dong et al., 2025) and CHRN (Gao et al., 2023), optimize feature smoothness and neighborhood consistency. Although effective within their respective domains, these methods generally lack cross-domain generalization. Models trained on one graph (e.g., citation network) often fail catastrophically on another (e.g., social network) due to substantial differences in graph structure and feature semantics.

## A.2. Generalist Graph Anomaly Detection

To overcome the limitations of domain-specific models, recent research has pivoted towards Generalist Graph Anomaly Detection (GAD), aiming to establish "one-for-all" frameworks that generalize to unseen graphs without retraining.

Pioneering this direction, ARC (Liu et al., 2024) formulates GAD as a few-shot problem via in-context learning, employing a smoothness-based alignment module to map diverse datasets into a unified latent space alongside an ego-neighbor residual graph encoder to capture relative anomaly patterns. Advancing the foundation model paradigm, AnomalyGFM (Qiao et al., 2025a) adopts a pre-train–fine-tune strategy, explicitly aligning node residuals with learnable prototypes to support zero-shot inference and prompt tuning. In a similar vein, UNPrompt (Niu et al., 2024) introduces a unified neighborhood prompt mechanism, leveraging latent node predictability as a generalized anomaly measure to standardize heterogeneous neighborhood information. Finally, IA-GGAD (Zhang et al., 2026) tackles feature space shift and graph structure shift through anomaly-driven invariant learning, extracting domain-invariant features and structural correspondences for robust zero-shot performance. However, simply unifying feature dimensions or prompting strategies is insufficient to bridge the complex semantic gaps between heterogeneous graph data, often resulting in suboptimal cross-domain transfer.

## A.3. Cross-Domain Representation Alignment

Cross-domain representation alignment is a fundamental concept in transfer learning and domain adaptation, designed to mitigate the distribution shift between source and target domains. Its primary objective is to map data from disparate domains into a shared latent space where their feature distributions are statistically aligned, thereby facilitating positive knowledge transfer (Pan & Yang, 2009). Classic alignment strategies generally fall into two categories: statistical moment matching and subspace alignment. The former minimizes statistical distance metrics, such as Maximum Mean Discrepancy (MMD) (Gretton et al., 2012), to reduce the discrepancy between domain distributions. The latter, subspace alignment (Fernando et al., 2013), focuses on aligning the basis vectors of source and target domains to create a unified feature space. In the context of graph learning, these techniques have been successfully adapted for tasks like node classification. For instance, methods such as UDAGCN (Wu et al., 2020) and CDNE (Shen et al., 2020) employ adversarial training (Ganin et al., 2016) or attention mechanisms to learn domain-invariant node embeddings, effectively handling structural heterogeneity.

However, in the specific realm of Generalist Graph Anomaly Detection, explicit representation alignment remains significantly under-explored. Since generalist models aim to generalize to completely unseen graphs, traditional alignment strategies that rely on joint optimization with specific target domain data are inapplicable. Consequently, most existing generalist models (Liu et al., 2024; Qiao et al., 2025a) implicitly assume that unifying feature dimensions via linear projection is sufficient for generalization. They often overlook the misalignment of underlying feature subspaces, which can make negative transfer when semantic gaps are large. Unlike previous approaches, our work introduces Relational Fingerprint Extraction, which transforms heterogeneous raw features into universal relational attributes. This strategy explicitly aligns semantic distributions across diverse graph domains, effectively mitigating negative transfer.

## B. Algorithm of REFI-GAD

---

**Algorithm 1** Pseudo-Code of the Proposed REFI-GAD

---

**Input:** Source-domain graphs $\{\mathcal{G}_s^1 \dots \mathcal{G}_s^{n_s}\}$ with labels;
    Target-domain graph $\mathcal{G}_t$ with support set;
    Model parameters $\Theta$.
**Output:** Trained generalist GAD model with parameters $\Theta^*$.
  1: **Step 1: Relational Fingerprint Extraction**
  2: **for** each graph $\mathcal{G} = (\mathcal{V}, \mathcal{E}, \mathbf{X})$ **do**
  3:     Compute contextual attributes NP, ND, and GD according to Eqs. (1)– (4);
  4:     Compute structural attributes degree d and clustering coefficient LC according to Eqs. (5) and (6);
  5:     Construct relational fingerprints $\mathbf{P} \in \mathbb{R}^{n \times 5}$ according to Eq. (7);
  6: **end for**
  7: **Step 2: Train Fingerprint-grounded Generalist GAD Model**
  8: **for** each training episode **do**
  9:     Sample support sets $\mathcal{S}_n$, $\mathcal{S}_a$, and query set $\mathcal{Q}$;
10:     **Domain-Shared Representation Encoding:**
11:     Encode node representations using the shared projector MLP and the context-aware Transformer according to Eqs. (8)– (10);
12:     **SNR-guided Domain-Adaptive Refinement:**
13:     Generate dimension weights based on dimension-wise SNR scores according to Eqs. (11)– (12);
14:     Encode node representations via the refinement projector MLP;
15:     **Anomaly Scoring:**
16:     Compute anomaly scores based on support-query similarity according to Eqs. (13)– (15);
17:     Update model parameters $\Theta$ by minimizing the BCE loss according to Eq. (16);
18: **end for**
    Return trained model parameters $\Theta^*$;
19: **Training-Free Inference on Target Domain**
20: Extract relational fingerprints of the target graph $\mathcal{G}_t$;
21: Encode node representations using the trained model with parameters $\Theta^*$;
22: Compute anomaly scores for target nodes.

---

## C. Complexity Analysis

We analyze the computational complexity of REFI-GAD, which consists of two stages: Relational Fingerprint (REFI) Extraction and Cross-Domain Adaptive Representation Learning.

In the REFI Extraction stage, relational fingerprints are constructed via contextual and structural pattern extraction followed by a rank-based transformation. The resulting time complexity is $\mathcal{O}(|\mathcal{E}|d + n \log n)$, where $n$ and $|\mathcal{E}|$ denote the numbers of nodes and edges, and $d$ is the raw feature dimension.

In the representation learning stage, the pre-computed fingerprints are processed by a Transformer-based encoder operating in a node-wise manner without recursive message passing. Each node interacts with a small support set of size $k$, leading to a complexity of $\mathcal{O}(nd'^2 + nkd')$, where $d'$ denotes the hidden embedding dimension.

## D. Experimental Details

### D.1. Datasets

We select 14 datasets from four distinct domains, including citation networks, social networks, e-commerce networks, and behavioral networks, to comprehensively evaluate the model's ability to capture diverse anomaly patterns. Such data diversity is essential for assessing the generalization capability of our model when adapting to unseen graph domains. To facilitate cross-domain evaluation, these datasets are further divided into two subsets, referred to as Group 1 and Group 2. The key statistics of all datasets are summarized in Table 5, followed by detailed descriptions of each dataset.

*Table 5.* The statistics of datasets organized by application domains.

| Dataset | #Nodes | #Edges | #Features | Avg. Degree | #Anomaly | %Anomaly |
|---------|--------|--------|-----------|-------------|----------|----------|
| Citation Networks | | | | | | |
| Cora | 2,708 | 11,604 | 1,433 | 4.29 | 150 | 5.54 |
| Citeseer | 3,327 | 10,154 | 3,703 | 3.05 | 150 | 4.51 |
| ACM | 16,484 | 147,866 | 8,337 | 8.97 | 597 | 3.62 |
| Pubmed | 19,717 | 92,846 | 500 | 4.71 | 600 | 3.04 |
| CS | 18,333 | 167,986 | 6,805 | 9.16 | 600 | 3.27 |
| Social Networks | | | | | | |
| BlogCatalog | 5,196 | 345,558 | 8,189 | 66.50 | 300 | 5.77 |
| Flickr | 7,575 | 482,602 | 12,047 | 63.71 | 450 | 5.94 |
| Facebook | 1,081 | 55,104 | 576 | 50.98 | 25 | 2.31 |
| E-commerce Networks | | | | | | |
| Amazon | 10,224 | 351,216 | 25 | 34.35 | 693 | 6.78 |
| YelpChi | 23,831 | 98,630 | 32 | 4.14 | 1,217 | 5.11 |
| Photo | 7,650 | 241,306 | 745 | 31.54 | 450 | 5.88 |
| Behavioral/Interaction Networks | | | | | | |
| Weibo | 8,405 | 407,963 | 400 | 48.54 | 868 | 10.33 |
| Questions | 48,921 | 153,540 | 301 | 3.14 | 1,460 | 2.98 |
| Reddit | 10,984 | 157,032 | 64 | 14.30 | 366 | 3.33 |

GROUP 1 DATASETS

- **Citeseer** (Sen et al., 2008) is a citation network where nodes represent scientific publications and edges denote citation relationships. Each node is associated with bag-of-words textual features. It is widely used as a benchmark for graph representation learning and anomaly detection under homophilic structures.

- **CS** (Coauthor-CS) (Shchur et al., 2018) is an academic collaboration network constructed from the Microsoft Academic Graph. Nodes correspond to authors, and edges indicate co-authorship relations. Node features are derived from paper metadata such as keywords or abstracts.

- **ACM** (Tang et al., 2008) is a paper citation network collected from the ArnetMiner database. It captures citation relationships among research papers from multiple computer science venues, with textual features describing paper content.

- **BlogCatalog** (Ding et al., 2019) is a social network where nodes represent bloggers and edges denote friendship relationships. Node attributes are constructed from user profile information, and the dataset is commonly adopted for social anomaly detection tasks.

- **Amazon** (Musical Instruments) (Rayana & Akoglu, 2015) is a user-review graph constructed from the Amazon Musical Instruments category, which is widely used for opinion spam and fraud detection. Nodes correspond to users, and edges represent co-review relations between users who have reviewed the same product. Each node is associated with 25 handcrafted behavioral features derived from review activities, with the task of identifying malicious users.

- **Photo** (Amazon Photo) (Shchur et al., 2018) is an Amazon co-purchase network for photography-related products. Nodes represent products, and edges connect items frequently bought together. Node features are bag-of-words vectors extracted from product reviews. This dataset is widely used as a benchmark for evaluating model robustness on e-commerce graphs.

- **Weibo** (Kumar et al., 2019) is a large-scale user interaction graph collected from the Chinese microblogging platform. Nodes represent users and edges indicate interaction behaviors such as reposting or commenting. It is commonly used for social spammer and abnormal user detection.

GROUP 2 DATASETS

- **Cora** (Sen et al., 2008) is a citation network of machine learning papers. Nodes correspond to publications and edges represent citation links. Each node is described by a sparse binary word vector.

- **Pubmed** (Sen et al., 2008) is a biomedical citation network where nodes denote scientific articles and edges correspond to citation relationships. Node features are TF-IDF weighted word vectors extracted from paper abstracts.

- **Flickr** (Tang & Liu, 2009) is a large-scale social network in which nodes represent users and edges indicate friendship relations. Node attributes are constructed from user-generated content and metadata, and the dataset is widely used for behavioral anomaly detection.

- **Facebook** (Xu et al., 2022) is a Facebook Page-Page network where nodes represent public pages and edges denote mutual likes. It is commonly used for detecting anomalous or irregular pages.

- **YelpChi** (McAuley & Leskovec, 2013) is a heterogeneous review network for fraud detection. Nodes include users and reviews, while edges model reviewing and interaction behaviors. The task focuses on identifying fraudulent users or spam reviews.

- **Questions** (Platonov et al., 2023) is an interaction network constructed from user–question behaviors. Nodes represent entities such as users or questions, and edges capture answering or commenting relationships. It is widely adopted for behavioral anomaly detection.

- **Reddit** (Kumar et al., 2019) is a post interaction graph where nodes represent posts and edges connect posts commented on by the same user. Node features are GloVe embeddings of post content, and the dataset serves as a benchmark for large-scale graph learning and anomaly detection.

## D.2. Baselines

CLASSICAL BASELINES

- **GCN** (Kipf & Welling, 2016) is a foundational semi-supervised graph neural network that learns node representations via spectral graph convolutions. In the context of GAD, it is typically employed as a backbone encoder to perform node classification or reconstruction.

- **GAT** (Veličković et al., 2017) introduces attention mechanisms to assign adaptive importance weights to neighbors during aggregation. This allows the model to filter out noisy neighbors for detecting anomalies.

SELF-SUPERVISED GAD METHODS

- **CoLA** (Liu et al., 2021) is a contrastive learning framework designed for graph anomaly detection. It generates positive and negative pairs between a target node and its local subgraph, employing a discriminator to measure agreement scores, where low scores indicate anomalies.

- **ANEMONE** (Jin et al., 2021) improves upon CoLA by introducing a multi-scale contrastive learning scheme. It captures anomaly patterns across different scales (e.g., patch-level and context-level) simultaneously to handle diverse anomalous behaviors more effectively.

- **SmoothGNN** (Dong et al., 2025) focuses on the smoothness assumption of graph signals. It identifies anomalies by measuring the deviation of node features after applying graph smoothing filters, exploiting the fact that anomalies often exhibit high-frequency irregularity.

SUPERVISED GAD METHODS

- **BWGNN** (Tang et al., 2022) addresses the "spectral anomaly" phenomenon where anomalies manifest in high-frequency bands. It employs beta wavelet filters to manage pass-band frequencies flexibly, capturing anomalies that standard low-pass GNNs might overlook.

- **BGNN** (Ivanov & Prokhorenkova, 2021) integrates Graph Neural Networks with Gradient Boosting Decision Trees (GBDT) into a unified framework, leveraging both graph structural information and the ability of GBDTs to handle complex, heterogeneous tabular features.

- **CHRN** (Gao et al., 2023) proposes a curvature-aware framework to mitigate the "camouflage" issue in graph anomaly detection. By leveraging Ricci curvature, it re-weights edges to distinguish between structural connections and anomalous interactions.

- **AHFAN** (Wang et al., 2025) is an adaptive high-frequency anomaly detection network. It dynamically adjusts its spectral filtering mechanisms to capture varying frequency patterns of anomalies, aiming to address the limitations of fixed-filter approaches.

- **GGAD** (Qiao et al., 2024) is a generative graph anomaly detection framework. It models the distribution of normal nodes by learning conditional generative patterns of graph structures and attributes, identifying anomalies based on their inconsistency with these patterns.

GENERALIST GAD METHODS

- **ARC** (Liu et al., 2024) is a generalist framework designed to transfer anomaly detection knowledge across domains. It unifies attribute, relation, and component information to learn domain-invariant patterns, enabling zero-shot or few-shot detection on unseen datasets.

- **AnomalyGFM** (Qiao et al., 2025a) leverages the capabilities of Graph Foundation Models (GFMs). It utilizes pre-trained large-scale graph models to extract universal representations and adapts them to specific anomaly detection tasks through lightweight tuning mechanisms.

- **UNPrompt** (Niu et al., 2024) introduces a prompt-based learning paradigm to unsupervised GAD. It designs specific graph prompts to align pre-training tasks with downstream detection objectives, allowing the model to detect anomalies without extensive fine-tuning.

- **IA-GGAD** (Zhang et al., 2026) emphasizes learning invariant anomaly patterns across graph distributions, aiming to enhance robustness and generalization under distribution shifts.

## D.3. Experimental Details

### D.3.1. METRICS

Following existing work (Tang et al., 2023; Qiao & Pang, 2023; Pang et al., 2021), we employ two popular and complementary metrics to comprehensively evaluate the detection performance: Area Under the Receiver Operating Characteristic Curve (**AUROC**) and Area Under the Precision-Recall Curve (**AUPRC**).

- **AUROC** measures the probability that a randomly selected anomalous node is ranked higher than a randomly selected normal node. It reflects the ranking quality of anomaly scores across all possible decision thresholds.

- **AUPRC** characterizes the relationship between precision and recall by summarizing performance across different confidence thresholds.

For both metrics, a higher value indicates superior performance. To ensure the reliability of the results, we perform five-fold cross-validation and report both the mean and standard deviation.

### D.3.2. IMPLEMENTATION PIPELINE

In our experiments, to ensure a fair and comprehensive evaluation under the generalist GAD setting, the 14 datasets are divided into two disjoint groups. Evaluation proceeds in two rounds: in the first round, models are trained on Group 1 and tested on each dataset in Group 2; in the second round, the roles are reversed, with Group 2 as the training set and Group 1 as the test set. This ensures that each dataset is evaluated as a strictly unseen target domain. For REFI-GAD, the hidden embedding dimension $d'$ is set to 64, and the encoder consists of 4 Transformer layers. The model is trained with a batch size

of 512. In each epoch, training nodes are randomly sampled from each source dataset with a fixed normal-to-anomaly ratio of 10:1. The support set size $k$ is set to 50 by default and kept consistent across all few-shot methods. For traditional GAD algorithms, PCA is applied to unify the feature dimensions of all datasets to 64, enabling evaluation in a generalist GAD scenario. These baselines are evaluated every 20 epochs, and the best performance observed during training is recorded as the final result. Hyperparameters for all comparison methods are determined via random search. Strictly adhering to the generalist GAD paradigm, we avoid dataset-specific hyperparameter tuning; instead, a single set of optimal hyperparameters is applied consistently across all testing datasets to authentically assess cross-domain generalization.

## E. Additional Experimental Results

### E.1. Comparison Experiments

In this subsection, we present a more comprehensive evaluation of the detection performance of REFI-GAD against baseline methods. For clarity and conciseness, the main text reports only the mean AUROC results. Here, we provide the complete experimental statistics, including the mean and standard deviation of both AUROC (Table 6) and AUPRC (Table 7), to enable a more thorough assessment of detection effectiveness and result stability.

First, the detailed AUROC results, including the mean and standard deviation over five independent runs, are reported in Table 6. Consistent with the findings in the main text, REFI-GAD shows stable and competitive performance across diverse graph domains, reflecting its strong generalization ability under unseen graphs.

Second, considering the extreme class imbalance in anomaly detection, we report the AUPRC results in Table 7. REFI-GAD achieves the best performance on 8 out of 14 datasets and ranks within the top three on 12 datasets, demonstrating consistently strong precision–recall behavior. In terms of overall performance, it attains an average rank of 2.14, outperforming the strongest baseline, ARC, which achieves an average rank of 3.43. These results indicate that REFI-GAD is more effective at controlling false positives under highly imbalanced detection scenarios.

### E.2. Ablation Study

To validate the contribution of each component, we conduct a comprehensive ablation study on all 14 datasets, with the complete results reported in Table 8 and 9.

**Ablation Study of Key Components**. The relational fingerprint serves as the core component of our framework. Replacing it with standard PCA (w/o R) leads to a substantial average AUROC decrease of 20.39%, indicating that simple linear projections are insufficient for aligning heterogeneous semantic spaces. The Adaptive Refinement module (w/o D) is also critical; removing it results in an average AUROC drop of 0.70%, with a more pronounced decrease of 2.11% on the YelpChi dataset, highlighting its important role in identifying domain-specific anomaly-relevant dimensions. However, on the Amazon dataset, relational fingerprint construction resulted in a performance decline. This is because anomalies in this dataset are primarily manifested in raw feature discrepancies and exhibit low correlation with the graph structure. Nevertheless, our method still demonstrates substantial cross-domain generalization capabilities on this dataset, ranking third among all compared methods. Nevertheless, our method still demonstrates substantial cross-domain generalization capabilities on this dataset, ranking third among all compared methods.

**Ablation Study of Relational Attributes**. Ablating individual attributes (LC, d, NP, ND, GD) consistently degrades performance, verifying that no single attribute is redundant. Notably, removing the Neighborhood Property (w/o NP) results in the largest average decrease of **2.33%**, highlighting the critical importance of neighborhood context.

### E.3. Visualization Experiments

To intuitively evaluate cross-graph transferability, we visualize node embeddings on six datasets in Figure 7. As observed, baseline methods (ARC, IA-GGAD, UNPrompt) often produce entangled embeddings, with significant overlap between normal and anomalous nodes. In contrast, REFI-GAD consistently generates well-separated clusters. This improved separability indicates that the relational fingerprint effectively aligns heterogeneous features into a shared semantic space, while the domain-adaptive refinement highlights anomaly-relevant dimensions, enhancing discrimination between normal and anomalous nodes.

*Table 6.* Anomaly detection performance in terms of AUROC (%) ± std. Highlighted are the results ranked first, second, and third. The "Rank" column is recalculated based on the average ranking across datasets.

| Methods | Group 1 (Training on Group 2) | | | | | | | Rank |
|---|---|---|---|---|---|---|---|---|
| | Citeseer | CS | ACM | BlogCatalog | Amazon | Photo | Weibo | |
| GAD Methods | | | | | | | | |
| GCN | 48.32 ± 1.03 | 56.19 ± 1.23 | 50.43 ± 1.04 | 43.06 ± 0.92 | 58.69 ± 0.17 | 47.70 ± 2.10 | 46.40 ± 1.82 | 11.57 |
| GAT | 63.11 ± 1.92 | 59.25 ± 0.94 | 61.48 ± 0.94 | 60.50 ± 0.73 | 54.84 ± 2.68 | 45.63 ± 1.39 | 73.51 ± 2.66 | 9.14 |
| CoLA | 73.81 ± 2.87 | 65.99 ± 2.30 | 54.94 ± 4.14 | 60.58 ± 5.07 | 65.02 ± 10.86 | 65.18 ± 1.52 | 41.08 ± 7.06 | 8.00 |
| SmoothGNN | 90.42 ± 8.81 | 78.65 ± 5.25 | 78.00 ± 4.92 | 73.06 ± 1.91 | 50.75 ± 3.01 | 44.80 ± 5.02 | 85.96 ± 4.50 | 6.29 |
| ANEMONE | 41.36 ± 2.40 | 42.18 ± 1.19 | 45.53 ± 3.24 | 35.18 ± 3.43 | 43.51 ± 4.29 | 49.22 ± 4.87 | 35.57 ± 4.75 | 14.00 |
| AHFAN | 48.02 ± 1.74 | 59.09 ± 0.63 | 63.70 ± 1.21 | 59.60 ± 0.25 | 29.50 ± 2.60 | 52.16 ± 0.60 | 63.03 ± 5.92 | 10.71 |
| BWGNN | 60.91 ± 1.50 | 59.12 ± 1.92 | 61.80 ± 1.54 | 67.26 ± 1.55 | 50.40 ± 5.49 | 60.82 ± 2.63 | 81.62 ± 2.44 | 8.14 |
| BGNN | 55.14 ± 3.74 | 52.14 ± 0.86 | 52.01 ± 1.08 | 52.65 ± 1.09 | 52.17 ± 2.43 | 47.26 ± 0.87 | 77.47 ± 4.03 | 10.71 |
| CHRN | 74.25 ± 2.39 | 78.01 ± 0.75 | 76.88 ± 0.50 | 65.09 ± 0.51 | 52.93 ± 1.06 | 51.99 ± 1.19 | 57.77 ± 2.23 | 7.00 |
| GGAD | 67.03 ± 4.99 | 64.76 ± 2.20 | 67.02 ± 0.64 | 58.83 ± 0.07 | 50.36 ± 4.58 | 58.72 ± 1.78 | 70.23 ± 0.77 | 8.71 |
| Generalist GAD Methods | | | | | | | | |
| ARC | 91.59 ± 0.33 | 82.76 ± 0.08 | 79.87 ± 0.12 | 74.07 ± 0.20 | 79.08 ± 0.51 | 75.87 ± 0.16 | 85.54 ± 3.65 | 2.86 |
| AnomalyGFM | 50.44 ± 0.94 | 43.47 ± 0.74 | 36.98 ± 0.70 | 45.16 ± 1.50 | 48.62 ± 4.45 | 44.86 ± 1.51 | 43.16 ± 6.40 | 13.43 |
| UNPrompt | 71.86 ± 2.36 | 74.65 ± 1.27 | 73.92 ± 0.85 | 68.94 ± 0.14 | 65.08 ± 10.07 | 68.68 ± 3.27 | 47.21 ± 1.53 | 6.14 |
| IA-GGAD | 91.74 ± 0.31 | 92.94 ± 0.43 | 90.98 ± 0.40 | 75.22 ± 0.20 | 82.49 ± 1.66 | 69.45 ± 1.07 | 91.70 ± 0.23 | 2.00 |
| REFI-GAD | 96.62 ± 0.83 | 98.53 ± 0.28 | 95.54 ± 0.97 | 76.79 ± 1.29 | 71.21 ± 2.23 | 75.95 ± 1.59 | 92.66 ± 1.09 | 1.29 |

| Methods | Group 2 (Training on Group 1) | | | | | | | Rank |
|---|---|---|---|---|---|---|---|---|
| | Cora | Pubmed | Flickr | Facebook | YelpChi | Questions | Reddit | |
| GAD Methods | | | | | | | | |
| GCN | 32.41 ± 1.23 | 33.68 ± 0.65 | 38.07 ± 0.78 | 76.35 ± 0.56 | 51.21 ± 2.23 | 41.37 ± 0.67 | 46.80 ± 0.75 | 12.14 |
| GAT | 56.07 ± 1.08 | 67.21 ± 0.75 | 56.05 ± 1.30 | 55.33 ± 1.91 | 50.43 ± 2.08 | 57.37 ± 0.45 | 43.39 ± 0.41 | 10.71 |
| CoLA | 66.02 ± 2.06 | 70.09 ± 2.94 | 60.64 ± 2.84 | 70.88 ± 5.50 | 52.41 ± 1.55 | 53.23 ± 4.66 | 50.60 ± 1.27 | 8.57 |
| SmoothGNN | 88.39 ± 5.98 | 78.22 ± 6.62 | 76.95 ± 4.07 | 53.62 ± 8.98 | 61.30 ± 6.52 | 59.01 ± 4.83 | 54.09 ± 2.43 | 4.57 |
| ANEMONE | 47.08 ± 0.37 | 37.65 ± 0.61 | 37.07 ± 1.14 | 45.55 ± 3.60 | 50.88 ± 2.53 | 51.55 ± 1.16 | 52.41 ± 2.42 | 12.43 |
| AHFAN | 55.89 ± 1.16 | 64.08 ± 7.14 | 62.77 ± 8.24 | 32.11 ± 6.72 | 49.76 ± 2.34 | 54.01 ± 3.36 | 46.90 ± 0.36 | 11.43 |
| BWGNN | 63.62 ± 1.82 | 68.19 ± 0.62 | 68.37 ± 2.71 | 41.55 ± 2.56 | 52.76 ± 1.62 | 57.59 ± 3.49 | 60.24 ± 1.12 | 7.14 |
| BGNN | 52.66 ± 4.09 | 58.46 ± 2.59 | 52.56 ± 1.57 | 65.80 ± 14.26 | 50.67 ± 5.75 | 56.80 ± 2.62 | 41.80 ± 2.13 | 11.43 |
| CHRN | 68.18 ± 2.51 | 75.15 ± 1.30 | 64.32 ± 0.90 | 33.07 ± 2.16 | 51.72 ± 0.58 | 59.81 ± 1.44 | 53.45 ± 0.72 | 7.57 |
| GGAD | 71.15 ± 0.42 | 69.82 ± 0.20 | 59.79 ± 0.84 | 20.24 ± 0.73 | 52.14 ± 3.75 | 58.92 ± 0.86 | 53.78 ± 1.25 | 8.43 |
| Generalist GAD Methods | | | | | | | | |
| ARC | 87.91 ± 0.26 | 85.62 ± 0.25 | 74.53 ± 0.28 | 67.69 ± 0.89 | 53.34 ± 0.41 | 57.96 ± 0.16 | 60.94 ± 0.54 | 4.14 |
| AnomalyGFM | 43.94 ± 0.85 | 48.96 ± 1.19 | 47.67 ± 0.27 | 71.08 ± 4.15 | 46.21 ± 0.32 | 53.34 ± 0.49 | 54.71 ± 1.88 | 11.00 |
| UNPrompt | 69.56 ± 0.40 | 82.72 ± 0.96 | 69.60 ± 0.27 | 75.82 ± 5.98 | 53.91 ± 4.69 | 46.72 ± 1.00 | 55.51 ± 0.85 | 5.71 |
| IA-GGAD | 88.08 ± 0.58 | 88.07 ± 0.35 | 67.86 ± 0.96 | 79.76 ± 1.91 | 52.45 ± 0.67 | 59.13 ± 0.72 | 57.76 ± 1.14 | 3.71 |
| REFI-GAD | 96.14 ± 1.41 | 97.66 ± 0.40 | 89.61 ± 1.24 | 94.67 ± 1.27 | 83.40 ± 2.66 | 60.27 ± 1.80 | 62.11 ± 1.59 | 1.00 |

*Table 7.* Anomaly detection performance in terms of AUPRC (%) $\pm$ std. Highlighted are the results ranked **first**, second, and third. The "Rank" column is recalculated based on the average ranking across datasets.

| Methods | Group 1 (Training on Group 2) | | | | | | | Rank |
|---|---|---|---|---|---|---|---|---|
| | Citeseer | CS | ACM | BlogCatalog | Amazon | Photo | Weibo | |
| GAD Methods | | | | | | | | |
| GCN | $7.63 \pm 2.98$ | $8.35 \pm 3.53$ | $7.10 \pm 3.33$ | $4.95 \pm 0.16$ | $8.52 \pm 0.16$ | $5.48 \pm 0.39$ | $9.95 \pm 1.01$ | 10.14 |
| GAT | $7.37 \pm 0.61$ | $6.85 \pm 0.53$ | $8.06 \pm 0.55$ | $13.29 \pm 0.55$ | $7.95 \pm 0.45$ | $4.92 \pm 0.14$ | $53.35 \pm 3.83$ | 10.43 |
| CoLA | $15.34 \pm 2.30$ | $21.09 \pm 3.70$ | $18.16 \pm 2.39$ | $25.23 \pm 2.36$ | $12.99 \pm 4.17$ | $10.98 \pm 1.56$ | $24.57 \pm 6.31$ | 6.86 |
| SmoothGNN | $41.39 \pm 13.91$ | $17.06 \pm 3.50$ | $16.81 \pm 4.82$ | $20.33 \pm 2.75$ | $8.70 \pm 2.24$ | $5.16 \pm 0.73$ | $43.60 \pm 7.87$ | 7.71 |
| ANEMONE | $4.43 \pm 0.77$ | $3.18 \pm 0.25$ | $3.58 \pm 0.17$ | $4.54 \pm 0.28$ | $6.15 \pm 0.52$ | $6.76 \pm 0.53$ | $9.25 \pm 0.84$ | 13.14 |
| AHFAN | $45.16 \pm 2.64$ | $57.14 \pm 1.59$ | $61.29 \pm 1.05$ | $56.92 \pm 0.36$ | $28.43 \pm 5.27$ | $50.29 \pm 1.25$ | $57.48 \pm 7.19$ | 2.43 |
| BWGNN | $5.85 \pm 0.43$ | $4.09 \pm 0.35$ | $5.26 \pm 0.44$ | $10.24 \pm 0.75$ | $6.27 \pm 0.67$ | $7.97 \pm 0.56$ | $44.19 \pm 6.11$ | 11.29 |
| BGNN | $5.12 \pm 0.63$ | $4.45 \pm 0.56$ | $4.45 \pm 0.40$ | $6.42 \pm 0.18$ | $7.35 \pm 0.46$ | $5.16 \pm 0.11$ | $62.69 \pm 6.41$ | 12.00 |
| CHRN | $15.01 \pm 2.10$ | $15.17 \pm 3.57$ | $23.57 \pm 0.77$ | $14.76 \pm 0.84$ | $7.38 \pm 0.38$ | $6.02 \pm 0.29$ | $11.80 \pm 1.00$ | 8.57 |
| GGAD | $11.12 \pm 3.14$ | $5.64 \pm 0.43$ | $7.34 \pm 0.46$ | $7.67 \pm 0.02$ | $7.10 \pm 1.69$ | $7.47 \pm 0.45$ | $16.23 \pm 0.20$ | 10.86 |
| Generalist GAD Methods | | | | | | | | |
| ARC | $47.56 \pm 0.58$ | $36.39 \pm 0.14$ | $40.79 \pm 0.13$ | $35.17 \pm 0.22$ | $44.42 \pm 1.17$ | $30.25 \pm 1.11$ | $51.03 \pm 8.69$ | 3.14 |
| AnomalyGFM | $4.73 \pm 0.32$ | $2.74 \pm 0.11$ | $2.67 \pm 0.06$ | $5.17 \pm 0.23$ | $6.52 \pm 0.89$ | $5.32 \pm 0.45$ | $27.93 \pm 4.93$ | 12.71 |
| UNPrompt | $10.41 \pm 1.85$ | $18.30 \pm 3.06$ | $16.95 \pm 0.64$ | $28.90 \pm 1.79$ | $12.73 \pm 5.85$ | $16.34 \pm 5.96$ | $18.78 \pm 2.16$ | 6.71 |
| IA-GGAD | $48.82 \pm 0.68$ | $37.68 \pm 1.21$ | $47.82 \pm 0.45$ | $34.98 \pm 0.10$ | $50.79 \pm 2.90$ | $14.61 \pm 1.27$ | $71.61 \pm 0.28$ | 2.57 |
| REFI-GAD | $70.91 \pm 10.82$ | $70.46 \pm 2.46$ | $62.15 \pm 5.03$ | $19.42 \pm 2.37$ | $16.80 \pm 1.67$ | $22.73 \pm 3.96$ | $68.18 \pm 5.48$ | 2.71 |

| Methods | Group 2 (Training on Group 1) | | | | | | | Rank |
|---|---|---|---|---|---|---|---|---|
| | Cora | Pubmed | Flickr | Facebook | YelpChi | Questions | Reddit | |
| GAD Methods | | | | | | | | |
| GCN | $3.95 \pm 0.14$ | $2.18 \pm 0.06$ | $4.52 \pm 0.03$ | $4.95 \pm 0.20$ | $6.33 \pm 0.37$ | $2.47 \pm 0.05$ | $3.16 \pm 0.03$ | 11.86 |
| GAT | $6.28 \pm 0.35$ | $4.44 \pm 0.12$ | $12.19 \pm 0.78$ | $2.46 \pm 0.12$ | $6.14 \pm 0.57$ | $3.50 \pm 0.11$ | $2.67 \pm 0.02$ | 10.86 |
| CoLA | $10.97 \pm 1.13$ | $17.41 \pm 2.09$ | $23.03 \pm 3.50$ | $7.35 \pm 2.02$ | $5.67 \pm 0.14$ | $3.45 \pm 0.58$ | $3.53 \pm 0.22$ | 7.29 |
| SmoothGNN | $43.34 \pm 9.10$ | $13.28 \pm 3.21$ | $20.18 \pm 0.74$ | $4.57 \pm 4.33$ | $7.94 \pm 2.17$ | $4.69 \pm 0.78$ | $3.62 \pm 0.32$ | 5.86 |
| ANEMONE | $5.47 \pm 0.42$ | $2.48 \pm 0.06$ | $4.84 \pm 0.13$ | $2.57 \pm 0.52$ | $5.24 \pm 0.46$ | $3.16 \pm 0.06$ | $3.63 \pm 0.26$ | 11.57 |
| AHFAN | $6.25 \pm 0.12$ | $5.78 \pm 0.10$ | $15.12 \pm 0.32$ | $9.09 \pm 0.24$ | $6.10 \pm 0.17$ | $7.31 \pm 0.16$ | $2.95 \pm 0.12$ | 6.43 |
| BWGNN | $11.16 \pm 2.33$ | $7.33 \pm 1.06$ | $15.40 \pm 1.57$ | $2.19 \pm 0.43$ | $5.48 \pm 0.19$ | $4.30 \pm 0.42$ | $5.43 \pm 0.44$ | 7.71 |
| BGNN | $5.76 \pm 0.96$ | $3.58 \pm 0.28$ | $8.15 \pm 0.49$ | $7.09 \pm 3.75$ | $6.27 \pm 0.95$ | $3.84 \pm 0.69$ | $2.66 \pm 0.15$ | 8.57 |
| CHRN | $15.28 \pm 1.70$ | $10.38 \pm 0.62$ | $17.50 \pm 1.64$ | $1.70 \pm 0.07$ | $5.00 \pm 0.13$ | $4.50 \pm 0.46$ | $4.66 \pm 0.48$ | 7.43 |
| GGAD | $15.12 \pm 0.38$ | $5.95 \pm 0.04$ | $7.42 \pm 0.16$ | $1.41 \pm 0.01$ | $6.86 \pm 0.87$ | $4.32 \pm 0.06$ | $4.57 \pm 0.13$ | 8.71 |
| Generalist GAD Methods | | | | | | | | |
| ARC | $47.04 \pm 0.37$ | $32.14 \pm 0.52$ | $39.59 \pm 0.12$ | $7.04 \pm 1.69$ | $6.23 \pm 0.08$ | $4.16 \pm 0.09$ | $4.93 \pm 0.15$ | 3.71 |
| AnomalyGFM | $5.17 \pm 0.45$ | $3.37 \pm 0.75$ | $5.86 \pm 0.14$ | $6.42 \pm 2.61$ | $5.42 \pm 0.06$ | $3.17 \pm 0.05$ | $3.62 \pm 0.28$ | 9.43 |
| UNPrompt | $11.15 \pm 1.08$ | $14.14 \pm 0.94$ | $23.67 \pm 1.86$ | $8.09 \pm 2.96$ | $6.62 \pm 1.13$ | $2.97 \pm 0.17$ | $3.82 \pm 0.16$ | 6.29 |
| IA-GGAD | $46.89 \pm 1.07$ | $32.27 \pm 2.22$ | $37.26 \pm 0.14$ | $5.59 \pm 0.50$ | $5.98 \pm 0.19$ | $4.38 \pm 0.11$ | $4.56 \pm 0.19$ | 4.71 |
| REFI-GAD | $73.94 \pm 6.89$ | $54.47 \pm 12.15$ | $35.42 \pm 0.337$ | $29.76 \pm 6.16$ | $22.27 \pm 3.17$ | $4.55 \pm 0.30$ | $5.60 \pm 0.40$ | 1.57 |

*Table 8.* Ablation study of REFI-GAD with different component combinations (AUROC %).

| Method | Group 1 (Training on Group 2) | | | | | | | Group 2 (Training on Group 1) | | | | | | | Avg. |
|---|---|---|---|---|---|---|---|---|---|---|---|---|---|---|---|
| | Cite | CS | ACM | Blog | Amz | Photo | Weibo | Cora | Pubmed | Flickr | FB | Yelp | Quest | Reddit | |
| w/o R&D | 61.41 | 70.94 | 62.96 | 73.08 | **83.62** | 64.98 | 79.12 | 52.80 | 66.75 | 73.41 | 55.04 | 55.55 | 55.23 | 58.27 | 65.23 |
| w/o R | 60.88 | 71.26 | 63.37 | 72.27 | 81.78 | 65.91 | 75.00 | 53.28 | 66.26 | 71.71 | 54.88 | 54.15 | 56.24 | 58.66 | 64.69 |
| w/o D | 95.95 | 98.42 | 94.64 | **76.79** | 70.39 | 75.90 | 91.57 | 96.07 | 97.33 | 88.08 | 93.30 | 81.29 | 60.16 | 61.48 | 84.38 |
| **REFI-GAD** | **96.62** | **98.53** | **95.54** | **76.79** | 71.21 | **75.95** | **92.66** | **96.14** | **97.66** | **89.61** | **94.67** | **83.40** | **60.27** | **62.11** | **85.08** |

*Table 9.* Ablation study of REFI-GAD with different relational attributes (AUROC %).

| Method | Group 1 (Training on Group 2) | | | | | | | Group 2 (Training on Group 1) | | | | | | | Avg. |
|---|---|---|---|---|---|---|---|---|---|---|---|---|---|---|---|
| | Cite | CS | ACM | Blog | Amz | Photo | Weibo | Cora | Pubmed | Flickr | FB | Yelp | Quest | Reddit | |
| w/o LC | 95.48 | 98.64 | 93.07 | 75.53 | 65.86 | 75.73 | 94.01 | 94.70 | 96.08 | 88.23 | 92.59 | 79.28 | 60.46 | 59.74 | 83.53 |
| w/o d | 95.97 | 98.35 | 94.58 | 77.13 | 69.64 | **78.53** | 93.30 | 96.08 | 96.56 | 79.78 | 88.65 | 81.44 | **61.62** | 60.89 | 83.75 |
| w/o NP | 94.05 | 97.12 | 90.97 | 71.39 | **72.24** | 72.59 | 82.40 | 94.23 | 95.12 | 89.53 | 94.92 | 81.93 | 60.05 | 62.02 | 82.75 |
| w/o ND | 96.12 | 96.26 | 92.98 | 78.54 | 70.19 | 73.33 | **94.89** | 93.93 | 97.12 | 85.85 | **95.96** | 82.14 | 59.70 | 60.53 | 84.11 |
| w/o GD | **96.84** | **98.89** | **95.57** | **81.47** | 70.46 | 68.00 | 92.30 | **96.62** | 97.62 | 82.05 | 88.56 | 80.37 | 58.93 | 59.46 | 83.37 |
| **REFI-GAD** | 96.62 | 98.53 | 95.54 | 76.79 | 71.21 | 75.95 | 92.66 | 96.14 | **97.66** | **89.61** | 94.67 | **83.40** | 60.27 | **62.11** | **85.08** |

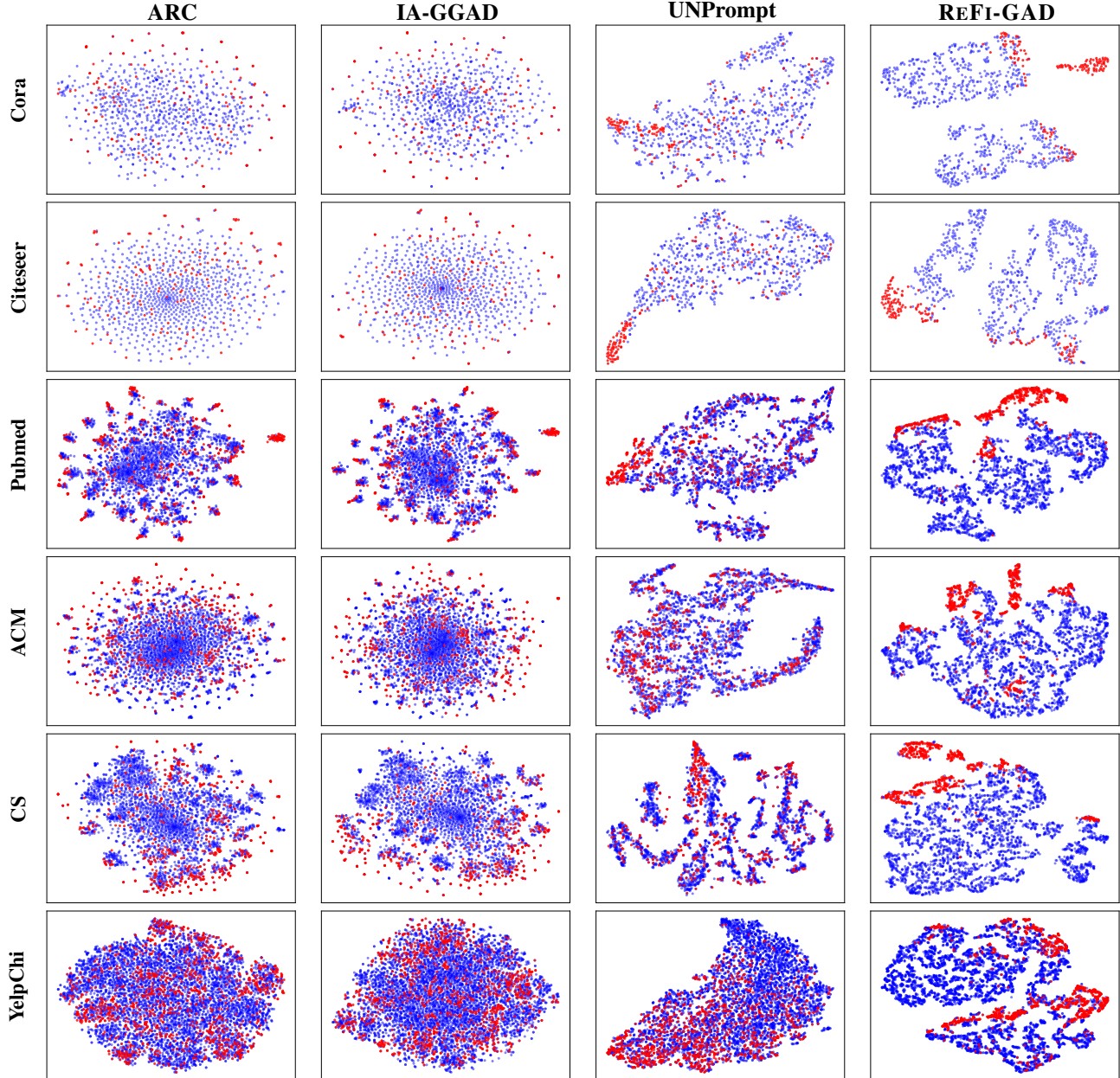

*Figure 7.* Visualization of node embeddings on six datasets. The results on Cora, Citeseer, Pubmed, ACM, CS, and YelpChi consistently show that our method achieves clearer separation between normal (blue) and anomalous (red) nodes.

