# OpenReview forum: "Rethinking Feature Alignment in Generalist Graph Anomaly Detection: A Relational Fingerprint-based Approach"
_ICML.cc/2026/Conference — ICML 2026 regular_

### Official Review · Reviewer_my47 · 2026-03-10

**Soundness:** 4
**Presentation:** 3
**Significance:** 3
**Originality:** 4
**Overall Recommendation:** 5
**Confidence:** 4

**Summary:**

This paper addresses the negative transfer problem that existing generalist GAD models encounter in practical tasks. The authors propose a novel feature alignment strategy that constructs aligned features by measuring the anomaly-indicative signals of nodes, thereby effectively achieving a dual unification of feature semantics and dimensionality across diverse graph datasets. Furthermore, an SNR-based domain adaptive module is introduced to further enhance the model's generalization performance. Extensive experiments demonstrate that the proposed algorithm exhibits significant performance advantages in cross-domain transfer scenarios.

**Compliance With Llm Reviewing Policy:**

Affirmed.

**Final Justification:**

The authors' feedback has addressed my concerns, thus increasing my confidence in the evaluation. I believe the model is of practical value in its compatibility with large-scale/memory-constrained scenarios and training without outliers.

**Key Questions For Authors:**

1. Please clarify whether the traditional supervised GAD baselines utilized the same few-shot support set in the target domain for fine-tuning during evaluation. If not, it is highly recommended to add these experiments to ensure a fair comparison.

2. The authors should provide an analysis of the algorithm's space complexity.

3. Could the authors discuss the extent to which their algorithm relies on known anomalous samples in the target domain? Furthermore, would the model fail to work if only normal nodes (a one-class setting) are provided as the support set in the target domain?

**Limitations:**

yes

**Strengths And Weaknesses:**

Strengths

1. The paper exhibits strong technical novelty. Unlike existing generalist GAD algorithms that primarily focus on detection mechanisms, this paper focuses on the more fundamental issue of effectively achieving cross-domain feature alignment across graph datasets, thereby exploring a novel technical route.
2. The method is well-designed with a clear logical flow. The paper provides a detailed and insightful analysis of the design rationale behind the core "relational fingerprint construction" module, allowing readers to easily grasp the primary focus of the research.
3. The experimental evaluation is extensive. The authors not only verify the superiority of their proposed algorithm through broad comparative experiments but also thoroughly investigate the specific contributions of each component via extensive ablation and visualization studies.

Weaknesses

1. The proposed method utilizes a few-shot support set containing ground-truth labels in the target domain, but it is unclear whether the compared traditional supervised baselines (such as BWGNN and AHFAN) were also permitted to use this target-domain supervisory information for fine-tuning.
2. Although the paper provides a time complexity analysis for the proposed algorithm, it lacks an analysis of the space complexity. Given that the construction of the relational fingerprint appears to require storing a substantial amount of matrix information, it is necessary to discuss the space complexity.
3. In real-world applications, it may be difficult to obtain anomalous samples to construct a support set in certain domains. However, this paper does not explore the model's performance in scenarios where only normal nodes are provided as the support set in the target domain.

---

> ### Author Rebuttal · Authors · 2026-03-31
>
> $\textbf{Q1: }$Please clarify whether the traditional supervised GAD baselines utilized the same few-shot support set in the target domain for fine-tuning during evaluation. If not, it is highly recommended to add these experiments to ensure a fair comparison.
>
> $\textbf{Response: }$We thank the reviewer for this question. To ensure a fair comparison, we conducted additional experiments on the Group 1 datasets, allowing traditional supervised methods (e.g., BWGNN and AHFAN) to leverage few-shot target-domain samples.
>
> The results show that with few-shot fine-tuning, BWGNN and AHFAN achieve average AUROC scores of 65.17% and 77.32% across seven datasets, improving over 63.13% and 53.59% without fine-tuning. In contrast, our method attains 86.76% without any parameter tuning, clearly outperforming these enhanced baselines and demonstrating its efficiency and superiority in few-shot cross-domain scenarios.
>
> $\textbf{Q2: }$The authors should provide an analysis of the algorithm's space complexity.
>
> $\textbf{Response: }$ReFi-GAD employs mini-batching and a memory-release mechanism to avoid dense global matrix formation. By condensing the graph data into a compact $N \times 5$ representation, the model maintains a linear space complexity $O(N)$ post-extraction. This efficiency enables scaling to graphs with tens of millions of edges, as evidenced by our strong results on Elliptic (70.53% AUROC) and T-finance (89.87% AUROC). These experiments confirm that ReFi-GAD is highly effective in large-scale, memory-constrained environments.
>
> $\textbf{Q3: }$Could the authors discuss the extent to which their algorithm relies on known anomalous samples in the target domain? Furthermore, would the model fail to work if only normal nodes (a one-class setting) are provided as the support set in the target domain?
>
> $\textbf{Response: }$We thank the reviewer for this question. We conducted additional experiments under the one-class setting: given 100 normal nodes as the support set, our method achieves an average AUROC of 79.46 across all datasets. Although this is lower than the setting using both normal and anomalous nodes, it still significantly outperforms the best baselines (ARC: 75.48, IA-GGAD: 77.69). This demonstrates that our method maintains strong anomaly detection capability even when relying solely on normal nodes.

---

> > ### Author Rebuttal · Reviewer_my47 · 2026-04-01
> >
> > The authors' feedback has addressed my concerns, thus increasing my confidence in the evaluation. I believe the model is of practical value in its compatibility with large-scale/memory-constrained scenarios and training without outliers.

---

> > > ### Author Response · Authors · 2026-04-04
> > >
> > > Thank you for the positive feedback. We are glad that our response addressed your concerns, and we will incorporate the discussed improvements into the final version.

---

### Official Review · Reviewer_pf56 · 2026-03-12

**Soundness:** 4
**Presentation:** 3
**Significance:** 4
**Originality:** 4
**Overall Recommendation:** 5
**Confidence:** 4

**Summary:**

This paper investigates the negative transfer issue commonly encountered by existing generalist GAD models in cross-domain scenarios. The authors propose a novel feature-alignment strategy that enables effective alignment across diverse graph datasets. Furthermore, they enhance the model’s domain adaptation capability through an SNR-guided domain-adaptive refinement mechanism. Extensive experimental results under multiple cross-domain settings demonstrate that the proposed method achieves superior performance compared to state-of-the-art approaches.

**Compliance With Llm Reviewing Policy:**

Affirmed.

**Key Questions For Authors:**

Q1: The paper uses rank transformation to standardize the scale of relational fingerprints. What is the specific advantage of this operation, and why not simply apply commonly used normalization methods such as min-max scaling?

Q2: When considering contextual patterns, the authors consider neighbourhood position and directional consistency as well as global directional consistency. Why is global positional consistency excluded from consideration?

Q3: After constructing the relational fingerprints, why do the authors choose not to use a GNN for graph learning, instead ignoring the graph structure and introducing a Transformer to capture inter-sample relationships? What are the advantages of this approach?

**Limitations:**

Yes.

**Strengths And Weaknesses:**

# Strengths

S1. The paper is well-structured and fluent, with the authors clearly presenting their research focus and methodological details.

S2. The authors identify the shortcomings of feature alignment strategies in existing GAD foundation models, and propose "Relational Fingerprints" to capture anomaly-indicative information of nodes. This achieves efficient cross-domain feature alignment for graph data.

S3. This paper introduces an SNR-guided domain-adaptive refinement, effectively enhancing the model's flexibility during cross-domain generalization.

S4. The authors verify the effectiveness of each proposed component through extensive experiments, proving that the methodology is rationally designed and its modules cooperate seamlessly.

S5. The comparative experiments are well-designed and sufficient, validating that the proposed method maintains significant performance advantages across different transfer scenarios.

# Weaknesses
W1. The relational attributes in the paper appear to be chosen empirically; providing a theoretical basis or guidance for their selection would strengthen the work.

W2. The train/test data allocation seems to be different to several previous studies. The authors should better clarify the advantages of using such a new allocation of two-group training/testing datasets.

W3. Some text in the figures is too small, such as the axis values in Fig. 4(b), and should be further optimized for clarity.

---

> ### Author Rebuttal · Authors · 2026-03-31
>
> $\textbf{W1: }$The relational attributes in the paper appear to be chosen empirically; providing a theoretical basis or guidance for their selection would strengthen the work.
>
> $\textbf{Response: }$We thank the reviewer for this suggestion. Our 5D relational fingerprint (ReFi) is theoretically motivated: contextual dimensions (NP, ND, GD) are based on the Homophily Hypothesis and statistical outlier principles, while structural dimensions (degree, local clustering) draw on Scale-Free Network Theory and the Triadic Closure Principle. Together, they serve as general indicators of graph anomalies rather than domain-specific heuristics. We will include these justifications and references in the final manuscript.
>
> $\textbf{W2: }$The train/test data allocation seems to be different to several previous studies. The authors should better clarify the advantages of using such a new allocation of two-group training/testing datasets.
>
> $\textbf{Response: }$Existing works often use a fixed dataset split, assigning some datasets as training and others as testing. However, this setup has two limitations: (1) it is hard to evaluate the model’s stability across different source domains, and (2) it cannot comprehensively assess the model’s cross-domain transferability, particularly its reverse transfer performance on datasets used for training.
>
> To address this, we alternate the two groups as training and testing sets, allowing each dataset to serve as both source and target. This provides a more comprehensive and objective evaluation of the model’s generalization and transferability.
>
> $\textbf{W3: }$Some text in the figures is too small, such as the axis values in Fig. 4(b), and should be further optimized for clarity.
>
> $\textbf{Response: }$Thank you for your suggestion. We will address this issue in the final version.
>
> $\textbf{Q1: }$The paper uses rank transformation to standardize the scale of relational fingerprints. What is the specific advantage of this operation, and why not simply apply commonly used normalization methods such as min-max scaling?
>
> $\textbf{Response: }$To validate our normalization strategy, we conducted additional ablation experiments. Removing rank normalization leads to an average AUROC drop of 3.26% (up to 13.96% on Photo), as unnormalized relational attributes exhibit large cross-domain magnitude variations that hinder transferability. Using standard $l_1$ normalization causes a drastic 35.31% drop, since topological features (e.g., node degree) typically follow long-tailed distributions, and conventional methods over-smooth extremes, weakening discriminative power. In contrast, rank normalization models relative node positions, effectively eliminating scale disparities while preserving structural ordering, ensuring superior cross-domain transferability. These results will be included in the final manuscript.
>
> $\textbf{Q2: }$When considering contextual patterns, the authors consider neighbourhood position and directional consistency as well as global directional consistency. Why is global positional consistency excluded from consideration?
>
> $\textbf{Response: }$The reason for not considering global positional consistency is that positional consistency attributes capture a node’s relative relationships within its local structure, which are highly localized and dependent on the data distribution. Their global statistics lack stable references across datasets, making them less transferable. In contrast, directional consistency reflects relative trends (e.g., ordering relationships) and exhibits better scale invariance and distributional stability, making it suitable for global modelling. Based on this, we apply global consistency only to directional features to enhance cross-domain generalization.
>
> $\textbf{Q3: }$After constructing the relational fingerprints, why do the authors choose not to use a GNN for graph learning, instead ignoring the graph structure and introducing a Transformer to capture inter-sample relationships? What are the advantages of this approach?
>
> $\textbf{Response: }$The proposed relational fingerprints explicitly encode rich graph structural information. Therefore, the model does not rely on a GNN for structural modelling. Instead, we introduce a Transformer to capture higher-order relationships and shared anomaly patterns among samples from a global perspective.

---

> > ### Author Rebuttal · Reviewer_pf56 · 2026-04-02
> >
> > Thanks for the response. My main concerns have been well addressed.

---

> > > ### Author Response · Authors · 2026-04-04
> > >
> > > Thank you for the positive feedback. We are glad that our response addressed your concerns, and we will incorporate the discussed improvements into the final version.

---

### Official Review · Reviewer_J8N3 · 2026-03-13

**Soundness:** 2
**Presentation:** 3
**Significance:** 2
**Originality:** 2
**Overall Recommendation:** 4
**Confidence:** 4

**Summary:**

This paper addresses the problem of generalist graph anomaly detection (GAD), which aims to learn a universal detector that can identify anomalies on unseen graphs without graph-specific retraining. The authors proposed a new method named REFI-GAD, which is based on a critical limitation in existing approaches, i.e., while methods like PCA-based projection align feature dimensions across domains, they fail to achieve semantic alignment, leading to widespread negative transfer. The method introduces a Relational Fingerprint which consists of a 5-dimensional universal representation capturing anomaly-indicative patterns from both contextual (neighborhood positional/directional consistency, global directional consistency) and structural (degree, local clustering coefficient) perspectives. Then the method employs rank-based transformation to ensure cross-domain comparability and combines a Transformer-based encoder for learning domain-invariant representations with an SNR-guided refinement module for domain-specific adaptation. Experiments on 10+ benchmark datasets demonstrate the effectiveness of the proposed method compared to SOTA.

**Compliance With Llm Reviewing Policy:**

Affirmed.

**Final Justification:**

The replies from the authors have addressed my concerns. I would like to increase the rating.

**Key Questions For Authors:**

1. Experiments on large-scale graphs
2. Theoretical analysis of the relational fingerprint
3. A minor question: since the 5-dim fingerprint will be the input of a neural network, what the performance will be if no ranking step is included?

**Limitations:**

yes

**Strengths And Weaknesses:**

**Strengths**
+ This method is well-motivated. The paper provides compelling empirical evidence that existing generalist GAD methods suffer from negative transfer despite large-scale pre-training. The insight leads to the proposed method that is based on the relational fingerprint solution.
+ The experiments are thorough and convincing, covering 14 diverse datasets across multiple domains (citation, social, e-commerce, behavioral networks). The ablation studies validate each component, and the visualization experiments provide intuitive evidence of improved separability.
+ REFI-GAD achieves SOTA performance with an average improvement of 7.39% over the strongest baseline (IA-GGAD) and demonstrates positive transfer on most datasets, addressing the core problem that existing methods exhibit negative transfer on most datasets. This validates the effectiveness of the technical design.

**Weakness**
- Limited theoretical analysis: While the paper provides strong empirical evidence, it lacks the theoretical analysis of why the proposed 5-dimensional fingerprint should be sufficient or optimal for capturing anomaly patterns across diverse domains. There's no formal guarantee that these five attributes comprehensively cover all (or the most common) anomaly-indicative patterns, nor any theoretical analysis of the convergence properties of the proposed optimization.
- Manual feature engineering concerns: The REFI design relies on manually designed relational attributes (NP, ND, GD, d, LC). While the authors acknowledge this limitation in the conclusion, it still raises questions about: (a) whether these five attributes generalize to all graph types and anomaly patterns, (b) whether important relational patterns might be missing, and (c) scalability to evolving anomaly types. The paper would benefit from an analysis of how to systematically determine which relational attributes are needed for a given domain.
- Computational complexity and scalability concerns: Although the paper provides computational complexity in the appendix, which shows the efficiency of the proposed method, the experiments only cover relatively small datasets (the largest appears to be ~50k nodes), and the discussion of scalability to large graphs is missing. In addition, a question about the complexity remains: does it include each of the fingerprint computation (it seems that some dimension requires pairwise calculation)?

---

> ### Author Rebuttal · Authors · 2026-03-31
>
> $\textbf{Q1: }$Experiments on large-scale graphs
>
> \textbf{Response: } To validate scalability, we evaluated our model on two ultra-large-scale benchmarks: Elliptic (\~204K nodes, \~234K edges) and T-finance (\~39K nodes, \~21.2M edges), using Group 1 and Group 2 as source domains, respectively. ReFi-GAD achieved average AUROCs of 70.53% and 89.87%, outperforming ARC (40.16% and 76.12%). These results demonstrate that the proposed relational fingerprints remain highly discriminative even in large and structurally complex graphs. We will include these results in the final manuscript.
>
> $\textbf{Q2: }$Theoretical analysis of the relational fingerprint
>
> $\textbf{Response: }$ We sincerely thank the reviewer for this insightful suggestion. The design of our 5D relational fingerprint (ReFi) is theoretically grounded, ensuring principled coverage of both contextual and structural anomalies [1].
>
> Contextual Dimensions (NP, ND, GD): The contextual dimensions are designed to capture deviations from local and global consistency. While the specific metrics are defined by our method, they are theoretically motivated by the Homophily Hypothesis [2] and the general principle of statistical outliers [3], which justify the detection of nodes that deviate from local or global patterns.
>
> Structural Dimensions (d, LC): Degree (d) is motivated by Scale-Free Network Theory [4], highlighting structural deviations of hubs or isolates. Local Clustering Coefficient (LC) leverages the Triadic Closure Principle [5], identifying micro-topological irregularities common in anomalous structures (e.g., fraud rings).
> Together, these dimensions serve as universal indicators of graph anomalies rather than domain-specific heuristics. We will explicitly integrate these theoretical justifications and include the corresponding references in the main text of the final manuscript.
>
> $\textbf{References: }$
>
> [1] Chandola, V., Banerjee, A., & Kumar, V. "Anomaly detection: A survey." ACM Computing Surveys, 2009.
>
> [2] McPherson, M., et al. "Birds of a feather: Homophily in social networks." Annual Review of Sociology, 2001.
>
> [3] Aggarwal, C. C. Outlier Analysis. Springer International Publishing, 2017.
>
> [4] Barabási, A. L., & Albert, R. "Emergence of scaling in random networks." Science, 1999.
>
> [5] Easley, D., & Kleinberg, J. Networks, Crowds, and Markets: Reasoning about a Highly Connected World. Cambridge University Press, 2010.
>
> $\textbf{Q3: }$A minor question: since the 5-dim fingerprint will be the input of a neural network, what the performance will be if no ranking step is included?
>
> $\textbf{Response: }$ We conducted ablation studies to evaluate the effect of rank normalization. Removing it leads to an average AUROC drop of 3.26% (up to 13.96% on Photo), as unnormalized relational attributes exhibit large cross-domain magnitude variations that hinder transferability. Using standard $l_1$ normalization causes a drastic 35.31% drop, since topological features (e.g., node degree) typically follow long-tailed distributions, and conventional methods over-smooth extremes, weakening discriminative power. In contrast, rank normalization models relative node positions, effectively eliminating scale disparities while preserving structural ordering, ensuring superior cross-domain transferability. These results will be included in the final manuscript.

---

> > ### Author Rebuttal · Reviewer_J8N3 · 2026-04-05
> >
> > Thanks for the reply. This clarifies my doubts.

---

> > > ### Author Response · Authors · 2026-04-05
> > >
> > > Thank you for the positive feedback. We are glad that our response addressed your concerns, and we will incorporate the discussed improvements into the final version.

---

### Official Review · Reviewer_B7PK · 2026-03-13

**Soundness:** 4
**Presentation:** 3
**Significance:** 3
**Originality:** 4
**Overall Recommendation:** 5
**Confidence:** 4

**Summary:**

This paper proposes a novel generalist graph anomaly detection model, whose core idea is to achieve cross-dataset feature alignment by constructing relational fingerprints for nodes. By capturing anomaly-indicative cues from both contextual and topological perspectives, the proposed strategy enables dual alignment in feature space and semantic space across graphs from different domains, thereby facilitating effective cross-domain transfer. Extensive experiments demonstrate that the proposed method exhibits superior cross-domain generalization performance compared to existing state-of-the-art approaches.

**Compliance With Llm Reviewing Policy:**

Affirmed.

**Key Questions For Authors:**

1. The SNR-guided Domain-Adaptive is applied to the node embeddings extracted by the Transformer. Why is it not performed directly on the five-dimensional relational fingerprint instead?

2. Appendix E.3 states that, on the Amazon dataset, anomalous signals are more strongly reflected in the raw features rather than in structural relations. Why does the proposed relational fingerprint not consider incorporating raw feature information?

**Limitations:**

Yes

**Strengths And Weaknesses:**

**Strengths**:

1. Clear research motivation. The authors identify that existing GAD foundation models commonly suffer from negative transfer in cross-domain generalization and empirically validate this issue through controlled experiments.

2. Strong novelty. Departing from conventional feature-alignment paradigms, the paper proposes a new alignment strategy that captures anomaly-indicative cues at the node level, enabling effective alignment in both the feature and semantic spaces across heterogeneous graph datasets.

3. Well-structured model design. All components are carefully designed around the core research question, with clearly defined functionalities, allowing readers to easily follow the overall methodological framework and design rationale.

4. Comprehensive experiments. Extensive experiments are conducted to support the authors’ claims and demonstrate the effectiveness of the proposed approach.


**Weaknesses**:

1. The manual-crafted 5-dimensional fingerprint required expert knowledge for design. The extensibility of the proposed methods on future new data could be limited, where new patterns may occur and need new fingerprint for data identification.

2. As shown in Fig. 1, the proposed method still suffers from negative transfer on two datasets; however, the authors do not provide a discussion of the possible causes.

3. The authors use t-SNE to show that the PCA-based dimensional alignment fails to unify feature semantics. However, no corresponding t-SNE visualization is provided for the relational-fingerprint-based alignment as a direct comparison.

---

> ### Author Rebuttal · Authors · 2026-03-30
>
> $\textbf{W1: }$The manual-crafted 5-dimensional fingerprint required expert knowledge for design. The extensibility of the proposed methods on future new data could be limited, where new patterns may occur and need new fingerprint for data identification.
>
> $\textbf{Response: }$ While the proposed 5D relational fingerprint is manually designed, it is not domain specific. It is grounded in the fundamental definition of anomalous nodes—deviations from regular patterns in local and global structures—capturing universal anomaly-indicative signals rather than dataset-specific heuristics. This design ensures the model focuses on shared patterns across datasets, enhancing extensibility to unseen data. Empirical results confirm strong cross-dataset generalization. Future work will explore incorporating domain-adaptive mechanisms for more complex or fine-grained anomalies.
>
> $\textbf{W2: }$As shown in Fig. 1, the proposed method still suffers from negative transfer on two datasets; however, the authors do not provide a discussion of the possible causes.
>
> $\textbf{Response: }$ We observe mild negative transfer on Amazon and Questions (−1.3% and −1.0%). This occurs because anomalies in these datasets are primarily feature-driven, while REFI emphasizes structural behaviour patterns. Despite this, our method ranks 1st on Questions and 3rd on Amazon, demonstrating robust cross-dataset transfer.
>
> $\textbf{W3: }$The authors use t-SNE to show that the PCA-based dimensional alignment fails to unify feature semantics. However, no corresponding t-SNE visualization is provided for the relational-fingerprint-based alignment as a direct comparison.
>
> $\textbf{Response: }$ We thank the reviewer for the suggestion. In the final version, we will include a t-SNE visualization after REFI alignment, as a direct comparison to Figure 2a, to more intuitively illustrate the alignment effect at the input level.
>
> $\textbf{Q1: }$The SNR-guided Domain-Adaptive is applied to the node embeddings extracted by the Transformer. Why is it not performed directly on the five-dimensional relational fingerprint instead?
>
> $\textbf{Response: }$ This design is motivated by two key considerations:
>
> First, the Transformer serves as a domain-shared encoder, aiming to extract domain-invariant interactions among relational attributes from the universal fingerprint. Performing domain adaptation directly on the five-dimensional input would bias the feature extraction process toward specific domains too early, thereby weakening the model’s ability to capture globally shared anomaly patterns.
>
> Second, graph anomalies are typically not characterized by a single dimension, but arise from the joint interplay of multiple relational attributes. Therefore, applying SNR-guided domain adaptation on the encoded representations enables finer-grained adaptation and more effectively enhances discriminative information.
>
> $\textbf{Q2: }$Appendix E.3 states that, on the Amazon dataset, anomalous signals are more strongly reflected in the raw features rather than in structural relations. Why does the proposed relational fingerprint not consider incorporating raw feature information?
>
> $\textbf{Response: }$ The primary reason is that feature patterns vary significantly across datasets, and naively incorporating raw features may harm cross-domain generalization. Nevertheless, our method still achieves 1st and 3rd place on the Questions and Amazon datasets, respectively, demonstrating strong overall competitiveness.
>
> In future work, we will further explore adaptive modelling of domain-specific anomalies within a general framework, and investigate how to more effectively incorporate raw feature information.

---

> > ### Author Rebuttal · Reviewer_B7PK · 2026-04-03
> >
> > Thank you for the rebuttal. It has cleared up my previous concerns, and I will be keeping my score as is.

---

> > > ### Author Response · Authors · 2026-04-04
> > >
> > > Thank you for the positive feedback. We are glad that our response addressed your concerns, and we will incorporate the discussed improvements into the final version.

---

### Decision · Program_Chairs · 2026-04-30

**Decision:**

Accept (regular)

**Comment:**

This paper investigates the problem of generalist graph anomaly detection and proposes a novel framework that leverages relational fingerprints to align semantic features and reduce negative transfer across domains. Abundant experiments also verify the superiority of the method. The reviewers raised some concerns regarding the novelty of the proposed method compared to the PCA-based alignment method, as well as pointing out the limitations in terms of hand-selected features and theory. However, these issues were largely clarified by the authors through additional enrichment analysis and experiments, and all reviewers showed positive evaluations.